# Pore evolution mechanisms during directed energy deposition additive manufacturing

Kai Zhang [1,2] ✉, Yunhui Chen [1,2,3,4], Sebastian Marussi [1,2], Xianqiang Fan [1,2], Maureen Fitzpatrick [1,3], Shishira Bhagavath[1,2], Marta Majkut [3], Bratislav Lukic[3], Kudakwashe Jakata[3,5], Alexander Rack [3], Martyn A. Jones[6], Junji Shinjo [7], Chinnapat Panwisawas [8], Chu Lun Alex Leung [1,2] & Peter D. Lee [1,2] ✉

Porosity in directed energy deposition (DED) deteriorates mechanical performances of components, limiting safety-critical applications. However, how pores arise and evolve in DED remains unclear. Here, we reveal pore evolution mechanisms during DED using in situ X-ray imaging and multi-physics modelling. We quantify five mechanisms contributing to pore formation, migration, pushing, growth, removal and entrapment: (i) bubbles from gas atomised powder enter the melt pool, and then migrate circularly or laterally; (ii) small bubbles can escape from the pool surface, or coalesce into larger bubbles, or be entrapped by solidification fronts; (iii) larger coalesced bubbles can remain in the pool for long periods, pushed by the solid/liquid interface; (iv) Marangoni surface shear flow overcomes buoyancy, keeping larger bubbles from popping out; and (v) once large bubbles reach critical sizes they escape from the pool surface or are trapped in DED tracks. These mechanisms can guide the development of pore minimisation strategies.

Directed energy deposition (DED)[1] is a promising layer-by-layer additive manufacturing (AM) technology that fabricates complex geometries for high-value-added products[2]. DED is also applied to repair applications, such as the repair of damaged turbine blades[3]. However, the industrialisation of the DED process for applications in automotive, marine, aerospace and biomedical fields has been limited by porosity introduced during the process, as porosity can be detrimental to a component's final mechanical performance, especially fatigue life[4].

Porosity is a common feature in DED-produced components and has been observed in various alloys[5–7], including titanium alloys[6,8,9], nickel-based superalloys[7,10] and aluminium alloys[11]. Porosity mainly consists of gas porosity and lack of fusion features, categorised by their formation mechanisms[12]. Gas porosity can originate from feedstock, entrapment of shielding gas[12], and the evolution of gases such as

hydrogen which are less soluble in the solid than the liquid metal[13,14]. A lack of fusion porosity can be formed due to insufficient energy input[15]. Porosity in DED is generally investigated with ex situ observation techniques including metallographic observation and X-ray computed tomography[16–19]. However, these techniques fail to capture either the phenomena by which pores form, or the dynamics of their growth and migration. To develop high-performance DED components with minimal porosity, it is necessary to gain a clear understanding of pore evolution and dynamics mechanisms using in situ observations.

Many in situ X-ray imaging studies have been conducted to investigate dynamic phenomena during solidification[13,20–25], including the molten pool behaviour in laser powder bed fusion (LPBF)[26–36], but only few have been performed on DED[5,6,10,37]. Pore formation was studied during LPBF by combining in situ synchrotron X-ray imaging and

[1]Department of Mechanical Engineering, University College London, London WC1E 7JE, UK. [2]Research Complex at Harwell, Harwell Campus, Didcot OX11 0FA, UK. [3]ESRF—The European Synchrotron, 38000 Grenoble, France. [4]School of Engineering, RMIT University, Melbourne, VIC 3000, Australia. [5]Diamond Light Source, Harwell Campus, Oxfordshire OX11 0DE, UK. [6]Rolls-Royce plc, PO Box 31 Derby DE24 8BJ, UK. [7]Next Generation Tatara Co-Creation Centre, Shimane University, Matsue 690-8504, Japan. [8]School of Engineering and Materials Science, Queen Mary University of London, London E1 4NS, UK. ✉e-mail: kai-zhang@ucl.ac.uk; peter.lee@ucl.ac.uk

multi-physics modelling, and it was found that the high thermo-capillary force can eliminate pores from the melt pool[31]. Pores were also found to be formed at the end of the scan vector during laser turning due to the formation and subsequent collapse of deep keyhole depressions, such that pockets of inert shielding gas are trapped by the solidification front[38]. Two studies systematically investigated pore formation during LPBF using high-speed X-ray imaging[30,33]. It was found that pore formation can be caused by a critical instability at the bottom of the keyhole[30]. However, this mechanism does not apply to the DED process which has a larger laser spot size and a lower energy density than LPBF. Hence DED is normally in conduction mode with no keyhole[2,39], has a much larger molten pool[40,41], and includes powder bombardment[42] which can contribute to different bubble evolution and melt pool dynamics. In DED, Wolff et al.[5] reported pore formation mechanisms as a result of powder delivery, keyhole dynamics, melt pool dynamics and shielding gas in Ti-6Al-4V using a piezo-driven powder delivery DED system; however, the energy density used was much greater than many industrial-scale DED builds with a keyhole formed, and hence some of the phenomena observed were more typical of the LPBF process. Therefore, there is a strong demand in getting results with industrial-relevant conditions and at a high temporal resolution to explain these pore mechanisms and physics involved in DED.

Similarly, there have been many multi-physics and multi-scale models of the LPBF process[43–48], but for DED models, there have been a relatively limited number of studies using high-fidelity multi-physics models[49–54]. Mostly, materials deposition and layer accumulation[51,52] and melt pool flow field[53], have been discussed. Additionally, Yang et al.[49] modelled the flow characteristics during DED, and ultrasound-assisted DED, using multi-physics modelling, coupled with high-speed optical imaging, but not X-ray. Two of the current authors developed a high-fidelity physics-based simulation to capture the chemical mixing between titanium and dissimilar refractory metals and its corresponding thermal-fluid characteristics during the DED[50]. However, these DED models did not include the formation, migration and release of pores, although models of these phenomena were well established in LPBF models and the wider field of solidification modelling[20,48,54,55]. For example, Li et al.[48] numerically investigated pore dynamics in LPBF such as coalescence and surface escape. This study is very suggestive of the pore dynamics and its effect on the product quality, but the melt pool scale was smaller (width of ~100 μm), and the bubble buoyancy effect and the temperature dependence of thermo-physical properties were not included.

In DED, where the melt pool is larger for the conduction mode under representative industrial conditions, the bubble size could be larger, and the effects of melt flow and buoyancy still remain unanswered. It is worth investigating since the buoyancy force is proportional to the cube of the bubble diameter. Furthermore, the effect of powder bombardment is particular to DED, which adds disturbance to the bubble dynamics. Therefore, the bubble dynamics in DED should be investigated comprehensively. In the numerical simulation, the melt pool velocity field information can be directly obtained, and the extraction of each specific effect could be possible.

One of the key issues is that these DED models were mainly validated with high-speed optical and thermal imaging results, and limited to the surface-based phenomena[49,56,57]. Importantly, these models benefit significantly from high-resolution and high-speed X-ray imaging experimental results to both determine the key physics to include and for validation. Therefore, it is critical to reveal the pore and melt pool dynamics in DED by combining high-fidelity multi-physics modelling and in situ X-ray imaging experiments.

In this work, we perform in situ high-speed synchrotron X-ray imaging (>20 kHz) to investigate pore evolution mechanisms during DED-AM. We quantify the pore behaviour including formation, coalescence, pushing, migration, escape and entrapment in the radiographs. We also quantify how these phenomena are correlated to key DED processing parameters. A multi-physics and high-fidelity modelling is applied to validate the hypothesised mechanisms including bubble migration, coalescence, pushing and escape. Our work contributes to an in-depth understanding of the DED additive manufacturing process, providing insights into how pore minimisation strategies may be developed.

## Results and discussion
### Pore behaviour in DED

In situ high-speed synchrotron X-ray imaging was used to observe pore dynamics and formation during the DED-AM. The experiment was performed using a Blown Powder Additive Manufacturing Process Replicator version II (BAMPR-II) on the ID19 beamline at the European Synchrotron Radiation Facility (ESRF) (details of the BAMPR II system and experimental set up can be found in 'Methods' and Supplementary Fig. 1 and references about BAMPR[6,9,10]). The powder is the gas atomised nickel-based superalloy RR1000 with a D50 size of 90 μm (see Supplementary Information for composition and size distribution in Supplementary Fig. 2 and Supplementary Table 1).

Based on the observations made using high-speed synchrotron radiography, pore behaviour can be divided into five stages: (1) pore formation; (2) bubble coalescence and growth; (3) solid/liquid interface pushing of large bubbles; (4) large bubble entrainment in the molten pool; and finally, (5) bubble escape or entrapment (see Fig. 1):

(1) Pore formation: pores are observed to be generated predominantly from the feedstock when using gas atomised powders. When these powders are atomised using argon gas, small bubbles of argon are entrained at the centre of many of the powder particles. These bubbles of argon are released into the molten pool when the powders melt. As shown in Fig. 1a and Supplementary Movie 1, at $t = t_0$, a powder particle, marked with a blue circle, hits the melt pool and is partially submerged. At $t = t_0 + 3$ ms, as the powder melts into the melt pool, and the argon bubble, marked with a yellow circle, is transferred into the pool (see schematic in Fig. 1b). The second largest source of pores is from the melting of the previous track (see Fig. 1g, i, where the pore, marked with purple, is released from the previous layer into the melt pool). As will be demonstrated later, the pores in the previous tracks are feedstock argon pores that have been transferred to the molten pool and then captured by the solidification front and frozen into the track.

(2) Bubble coalescence and growth: we observed that small bubbles can coalesce into larger ones ($t = t_0 + 146$ ms) (Fig. 1c). The bubbles formed by the coalescence of a couple of feedstock bubbles recirculate with the Marangoni flow in the melt pool and continue to grow by coalescing with more small bubbles (at $t = t_0 + 150$ ms, the bubble marked in a yellow circle). The bubble, marked with a yellow arrow and circle, is observed to migrate from the recirculating flow of the front to the back of the melt pool ($t = t_0 + 146$ ms), and then coalesce with the bubble in the back ($t = t_0 + 150$ ms). This large bubble at the rear of the molten pool is formed by the coalescence of tens of feedstock bubbles, and surprisingly remains relatively stable in the flow for relatively long periods of time (~0.5 s in this condition), growing and being released in a periodic cycle, as discussed later. Figure 1d shows the schematic of bubble circulation and bubble lateral movement. Quantification of their instantaneous circulation velocities is discussed later in the bubble migration section.

In Fig. 1a–d, the outward Marangoni flow is expected to occur in the melt pool, as the surface temperature is the highest under the laser than near the edge of the melt pool, so the liquid flows from this low surface energy area out to the colder edges (higher surface energy) to minimise the free energy. This creates a very fast strong surface flow outward from the laser, creating recirculation flow cells at the front and back[58]. Bubbles are observed to follow the outward Marangoni flow in the melt pool. Based on the 2D projections of the events

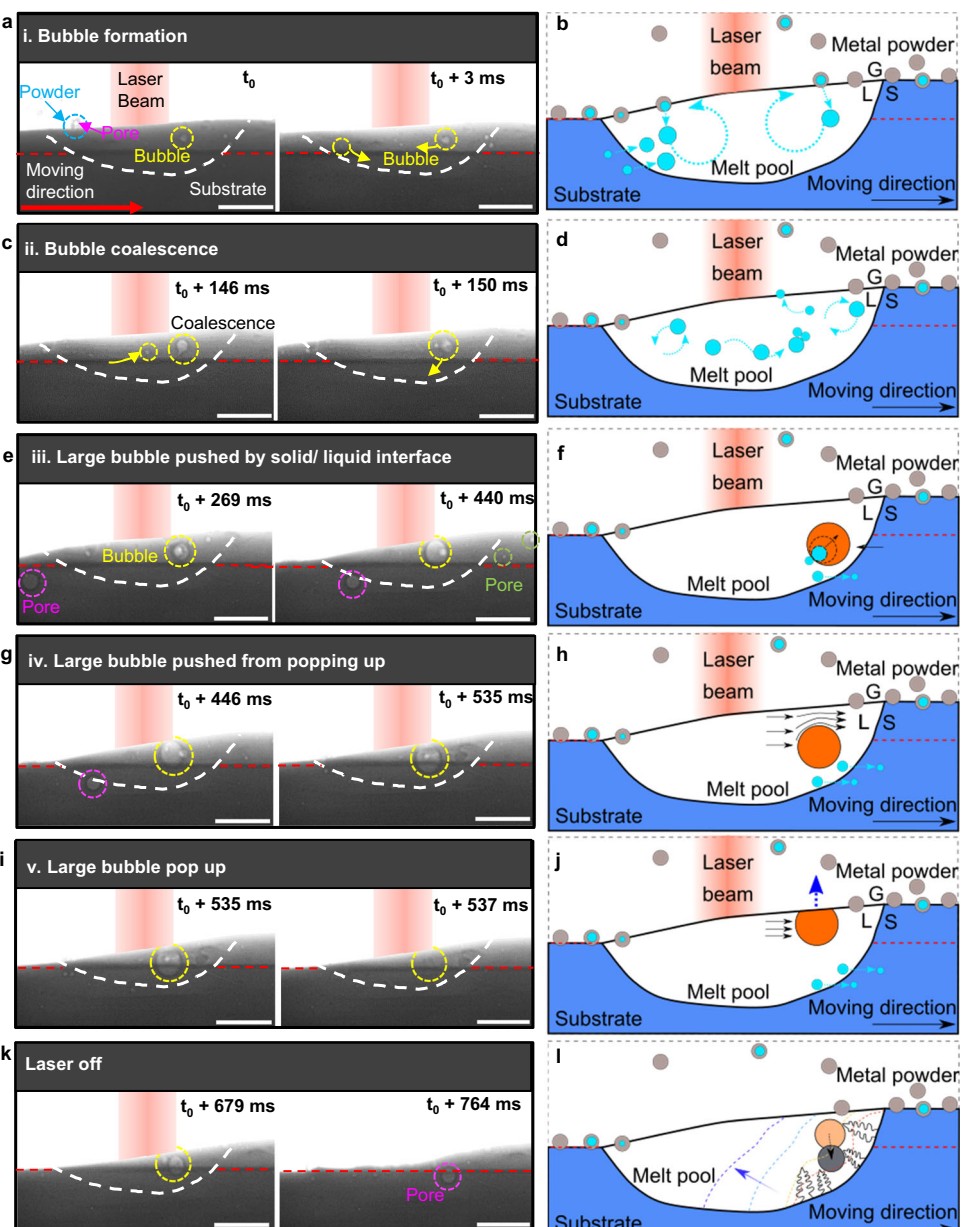

**Fig. 1 | Dynamic bubble behaviour and mechanisms during DED.**
**a**, **b** Radiographs with associated schematic showing a bubble formed from an argon pore inside a powder particle. Small bubbles are entrained in the recirculating flows in the melt pool. G represents gas, L represents liquid, S represents solid in the schematic. **c**, **d** Radiographs with associated schematic showing small bubbles coalescing into a larger bubble. Small bubbles often migrate from the front to the rear of the recirculating flows in the melt pool. **e**, **f** Radiographs with associated schematic showing a large bubble pushed by solid/liquid interface, growing as small bubbles coalesce into it. **g**, **h** Radiographs with associated schematic showing a large bubble entrained in the melt pool, prevented from bursting at the surface by the squeezed Marangoni shear flow. **i**, **j** Radiographs with associated schematic showing the large bubble (yellow circle) bursting at the melt pool surface after it reaches a critical size. **k**, **l** Radiographs with associated schematic showing the large bubble trapped by the solidification front when the laser is turned off. The substrate traverse speed is 2 mm s$^{-1}$, the laser power is 160 W, layer 1. The laser beam in the X-ray radiographs and corresponding schematics are shown in red colour, and the laser beam location is nearly symmetrical to the melt pool geometry, while it is slightly in the forward of the centre due to the advection of heat. See the video corresponding to (**a**, **c**, **e**, **g**, **i**, **k**) in Supplementary Movie 1. Scale bars in radiographs: 300 μm.

(Fig. 1c), in the front and back regions of the melt pool ($t = t_0 + 146$ ms and $t_0 + 150$ ms), bubbles are observed to recirculate, driven by the Marangoni flow.

(3) Solid/liquid interface pushing of bubbles: in situ radiography has been used to show that an advancing solid-liquid interface can either push or capture bubbles[13]; both pushing and capture mechanisms were observed here. For the smaller bubbles (25–40 μm), many were captured by the solidification front at the rear of the melt pool, forming pores in the track, such as the pores marked with green circles in Fig. 1e. However, the large coalesced bubbles are pushed by the

solidification front during the steady state, as shown in Fig. 1e, where a large bubble is pushed by the solidification front near the solid/liquid interface in the back of the melt pool (from $t = t_0 + 269$ ms to $t_0 + 440$ ms). This large bubble continues to grow as smaller bubbles flow from the rest of the pool and then coalesce with it. Surprisingly, this bubble remains in the melt pool, rather than rising under a strong buoyancy force.

(4) Large bubble entrainment in the molten pool: we observed that the large bubbles at the rear are pushed along ahead of the solidification front, and surprisingly do not rise to pop at the melt surface,

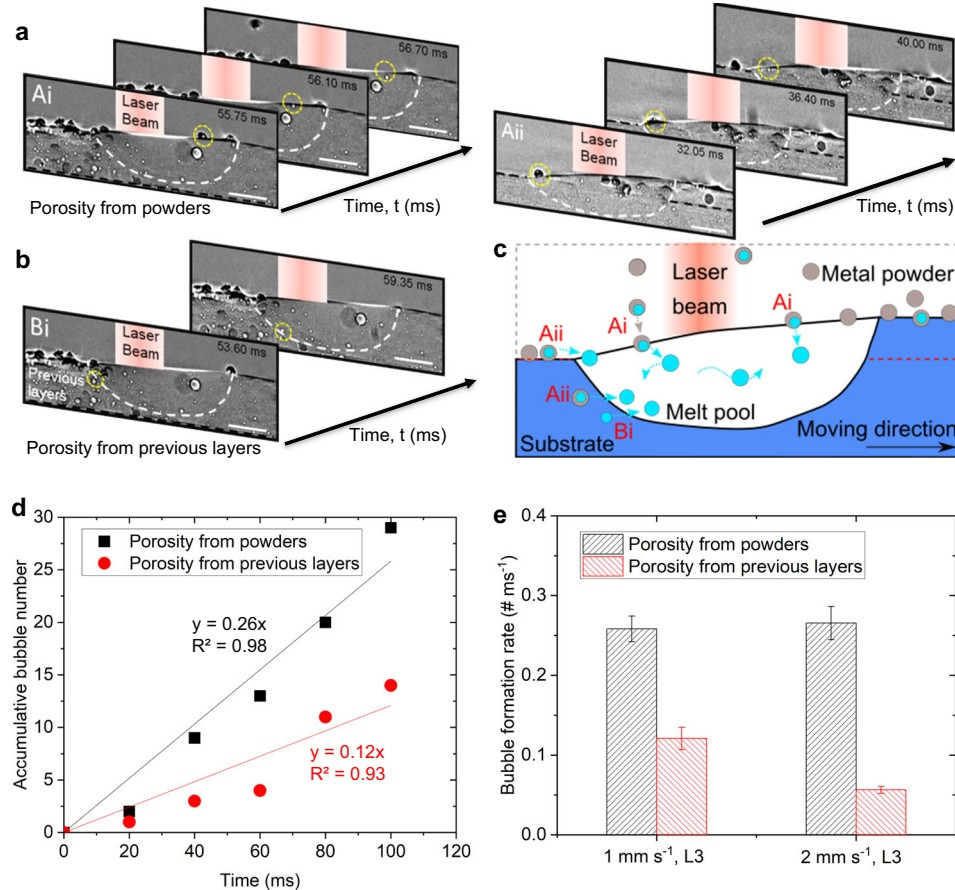

**Fig. 2 | Pore formation mechanisms during DED. a** Pore formation dynamics. Ai, a pore was captured to form in the melt pool from the porosity in the powder feedstock particle at a substrate traverse speed of 1 mm s$^{-1}$, a laser power of 160 W, layer 3; Aii, a pore formed from a powder when the laser melts the powders at a substrate traverse speed of 1 mm s$^{-1}$, a laser power of 160 W, layer 1. **b** Bi, a pore formed from the porosity in the previous layers. **c** Schematic illustration of the pore formation mechanism at a traverse speed of 1 mm s$^{-1}$, a laser power of 160 W, layer 3. **d** Accumulative number of pores from powders and previous layers with increasing time in DED at a traverse speed of 1 mm s$^{-1}$, a laser power of 160 W, layer 3. **e** Pore formation rate from porosity in powders and previous layers in DED at a traverse speed of 1 mm s$^{-1}$, a laser power of 160 W, layer 3; a traverse speed of 2 mm s$^{-1}$, a laser power of 160 W, layer 3, respectively. Error bars represent standard deviation. See the videos corresponding to (**a**–**c**) in Supplementary Movies 2 and 3. Scale bars in (**a**, **b**) are 300 μm.

remaining entrained in the molten pool (see bubble marked with a yellow circle in Fig. 1g). We hypothesise that the large pores are kept in the pool by the downward force exerted on them by the very fast-moving Marangoni surface shear flow that gets compressed above the bubble as it flows outwards over it (see schematic in Fig. 1h). The bubble appears to be maintained stably in the flow by the equal and opposite forces, until it exceeds a critical size.

(5) Bubble escape or entrapment in the solid: we also captured what happened to the bubbles in the end. Some bubbles escape from the melt pool. When the large bubbles grow beyond a critical size (~120 μm in diameter) by coalescence, the upward buoyancy force overcomes the downward Marangoni shear flow force, and the bubbles rise to the melt pool surface and burst, as shown in Fig. 1i, j. Using fast 20 kHz frame rate imaging (Supplementary Fig. 3), the large bubble escape process is clearly observed, namely, the large bubble moves close to the melt pool surface, coalesces with the melt pool surface and then bursts. Interestingly, many of the recirculating small bubbles burst as they reach the surface (detailed calculation can be found in the bubble escape section), perhaps due to the reduced blockage of the Marangoni shear flow as compared to larger bubbles.

As already discussed, the small bubbles are often entrapped in the solid-liquid interface, while the larger bubbles are usually pushed during the steady state. However, when the laser is turned off at the end of the track (Fig. 1k, l), both small and large bubbles are often captured by the solidification front towards the end of the track, as the

front becomes less planar (and often more dendritic as the thermal gradient reduces). This observation explains the propensity of large pores being found at the end of the track[59], a phenomenon confirmed by our tomography results (Supplementary Fig. 4).

These five stages of bubble behaviour depict the life cycle of bubbles in DED AM, and we observed that they repeat periodically during the building process, as discussed below in the 'Bubble growth' section.

## Pore formation mechanisms in DED-AM

We observed from the radiographs that pores mainly form from two sources. The first and dominant source is the gas atomised powder feedstock. Argon pores present in the powder feedstock are transferred into the molten pool as the powders melt. Figure 2a captures the phenomenon in detail as a powder particle hits the molten pool surface and the pore transfers into the melt pool after about 1 ms after the particle melts. Similar phenomena can be observed in the pore formation process Aii in Fig. 2a. The second source of porosity is the track material which is laid down on, initially a substrate machined from a DED-AM produced block, and after the first build, prior tracks. The substrate and prior tracks contain small pores that are clearly visible in the radiographs (Fig. 2a), and they are released into the melt pool when the laser beam remelts the substrate/prior track (Fig. 2b).

For the conditions used in this study, namely a gas atomised powder and conduction mode laser power, feedstock porosity is the

dominant source of porosity. This was quantified by counting the newly formed pores over 100 ms of the build for each source (Fig. 2d), with the argon pores in the feedstock powder introducing 2 to 4 times as many as pores enter from all other sources. The only other source of bubbles we observed was those entering from the prior tracks (Fig. 2d). However, ref. 5 suggested that during DED-AM of Ti-6Al-4V porosity can be generated from the feedstock, keyhole collapse, and by entraining gas when the powder particles enter the pool. For their conditions, using plasma atomised powders and laser conditions creating a keyhole, they concluded that feedstock porosity is a relatively insignificant contribution to the process with a contribution ratio of 0.22%[5]. Our results show that for the more industrial conditions used here, feedstock porosity becomes the major source of pores rather than a negligible one. This would be the one of major differences in pore formation between this work and the prior study[5].

The pore formation rate, defined here as the number of pores formed in the melt pool per millisecond, is shown in Fig. 2e. For the two build velocities used, this graph shows the pore formation rate from feedstock powder is 2 to 4 times higher than the bubble uptake from previous layers. It also shows the pore formation rate from powders is similar for both 1 and 2 mm s$^{-1}$, as expected since the powder feed rate, and hence the source of pores, is the same. However, a higher pore formation rate from porosity in previous layers is observed at 1 mm s$^{-1}$ than 2 mm s$^{-1}$. This is probably due to the smaller pool size at the

higher speed, and hence less remelted material entering the pool. Further, the porosity in the previous layers is greater at 1 mm s$^{-1}$.

Unlike the ref. 5, we did not observe any keyhole porosity in our experiment as we operated in a 'conduction' mode, with an energy density closer to industrial standards. Nor did we observe any pores formed from the delivery gas or entrained gas on particle bombardment. Powder particles are observed to gradually melt into the melt pool after they hit the melt pool surface and create ripples, see Supplementary Movies 2 and 3. Note, keyholes normally occur in the laser powder bed fusion process rather than DED, as LPBF is normally operated with a much higher laser power density[33]. Further discussions about the comparison between our work with the previous work[5] and the industrial DED can be found in Supplementary Discussion 1 and Supplementary Table 2.

## Bubble growth and pushing mechanisms

We carefully measured the bubble diameter changes against different processing conditions (including traverse speed, laser power and layers of build tracks) as an indication to understand the bubble growth mechanism, as shown in Fig. 3. Figure 3a shows the bubble diameter changes against different substrate traverse speeds. It is observed that the large bubble behaviour is periodic, with bubbles growing to a critical size and then escaping, with a new large bubble then forming in a similar location, repeating the process over the recorded distance of

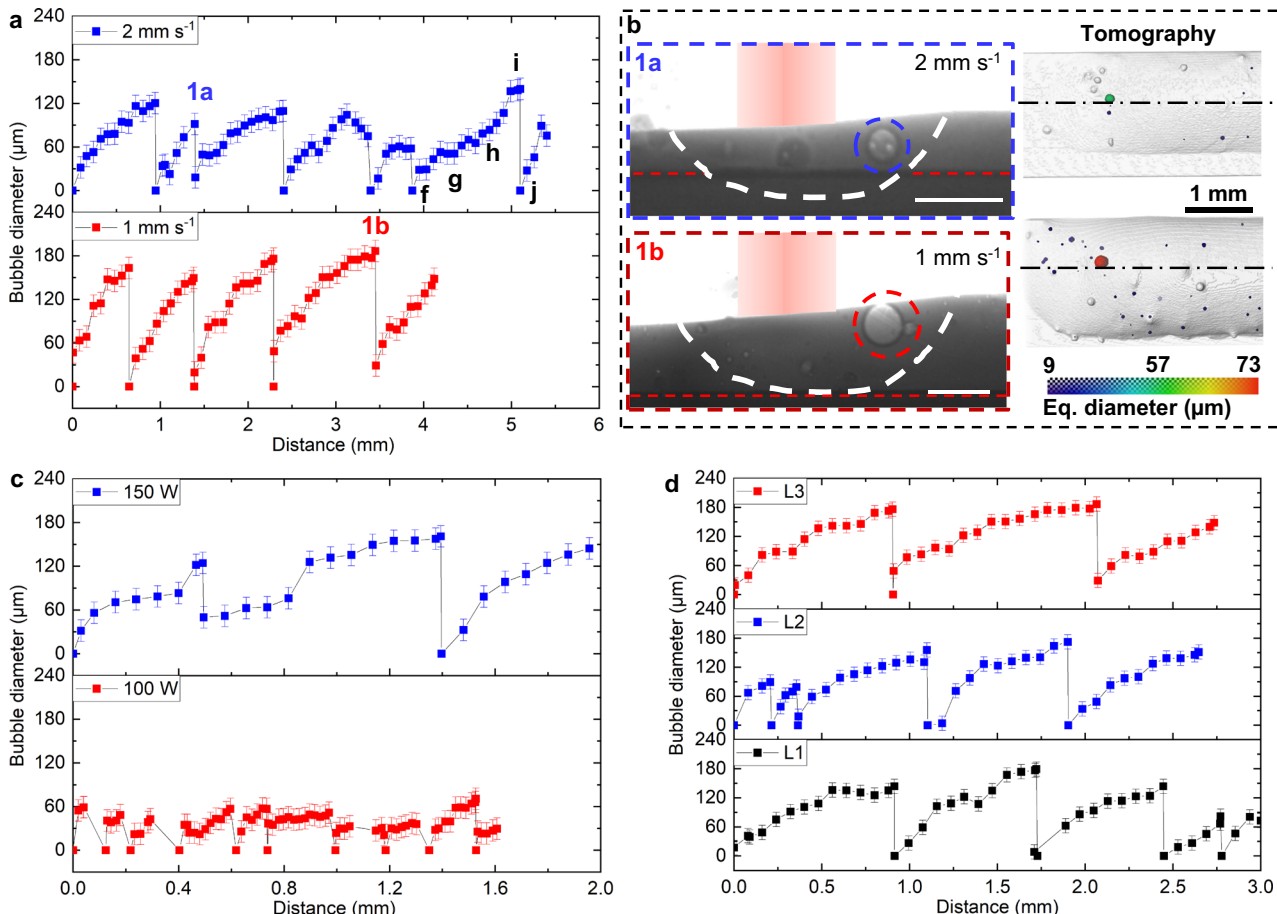

**Fig. 3 | Quantification of the bubble growth mechanisms. a** Bubble growth over different building parameters. Bubble diameter changes were tracked with moving distance over the building process in the third layer of the build for a substrate traverse speed of 2 and 1 mm s$^{-1}$, respectively. The laser power is 160 W. The bubble diameter error bars are calculated as ±2 pixels, equivalent to the segmentation uncertainty. **b** Radiograph examples at 2 and 1 mm s$^{-1}$ are shown in 1a and 1b (scale bars are 300 μm), with the corresponding tomographic rendered images overlaid

with the pore equivalent diameter. See the videos corresponding to (**b**) in Supplementary Movies 1 and 4. **c** Bubble growth over the building process in the first layer of build for a laser power of 150 and 100 W, respectively. The traverse speed is 1 mm s$^{-1}$. **d** Bubble growth over different layers. Bubble diameter changes were tracked over different layers of the build, namely, layers 1–3, the laser power is 160 W, and the traverse speed is 1 mm s$^{-1}$.

the build. The sudden diameter drops in Fig. 3a indicate the time when the large bubble escapes. The phenomena are compared for two different traverse speeds, 1 and 2 mm s⁻¹. We discovered that the maximum diameter a bubble can reach is about 180 μm at a traverse speed of 1 mm s⁻¹, which is larger than the condition at a traverse speed of 2 mm s⁻¹, where a maximum bubble diameter is measured to be about 120 μm. It is speculated that this is due to the changes in the Marangoni flow and buoyancy force in the larger and deeper melt pool at a traverse speed of 1 mm s⁻¹. At a traverse speed of 2 mm s⁻¹ the large bubbles travel approximately the same periodic distance (in about half the time before escaping) as compared to 1 mm s⁻¹. By counting the bubble number for coalescence, it is found that the average number of initial bubbles to coalesce the largest bubbles is over 30 at 2 mm s⁻¹ and over 70 at 1 mm s⁻¹, indicating that the bubble coalescence consumes a large number of bubbles. Since the largest bubbles volumes at 1 mm s⁻¹ were over double the largest bubbles volumes at 2 mm s⁻¹, they were probably formed by approximately double the number of smaller bubbles. The growth of the large pore provides convincing evidence of bubble coalescence, and although there may be some overlap of the bubble through the thickness, the high frame rate data shows small bubbles touch the larger pore and disappear, also providing strong evidence of coalescence, as shown in Supplementary Movie 3 at about $t = 41$ and 48 ms.

From the observation, we noted that melt pool size is an important factor for the bubble growth dynamics. The maximum bubble diameter is larger at 1 mm s⁻¹ than that at 2 mm s⁻¹. As shown in Fig. 3b, both the depth and width of the melt pool are larger at 1 mm s⁻¹, and the volume for bubble growth is larger, so the maximum bubble diameter is larger before it escapes.

The corresponding tomography results indicate that the maximum bubble diameter remains larger at 1 mm s⁻¹ when the laser is off (see Fig. 3b and the full build track in Supplementary Fig. 4). The large bubbles are kept in the back of the melt pool rather than other positions, this could be attributed to the different melt flow in various locations, as the melt flow can push the bubbles down in the back of the melt pool.

Figure 3c plots the bubble diameter with moving distance at a laser power of 150 W and 100 W. The maximum bubble diameter at 150 W is about 160 μm, which is more than two times of the bubble diameter of about 70 μm at 100 W. The larger maximum bubble diameter at a higher laser power could also be associated with the larger melt pool size for bubble growth. We also investigate the correlation between bubble behaviour and the different layers of build, as shown in Fig. 3d. From the results, we can confirm that the bubble behaviour, including the lifespan of the cycle and the maximum bubble diameter, is not affected by the differences of build layers. It was also found that the diameter of a large bubble is larger at a higher powder flow rate (Supplementary Fig. 5), this can be attributed to more argon pores entering the melt pool with a higher powder flow rate.

The melt pool size at different traverse speeds, laser powers and layers are plotted in Supplementary Fig. 6. The melt pool length and depth are both larger at a lower speed, higher laser power and greater powder feed rate, while the layer effect is insignificant. This is related to the bubble growth behaviour as shown in Fig. 3 and Supplementary Fig. 5, i.e., the larger melt pool allows the larger maximum bubble size reached.

The bubble pushing behaviour over different build conditions was investigated. We noted that bubbles were being pushed in the melt pool while growing for a certain distance before they escaped. We have discussed previously that the pushing behaviour is related to the combination of Marangoni flow and buoyancy force. And the time of bubbles being pushed equals to the lifespan of the bubbles discussed in this section and is closely related to the growth rate we measured. We hypothesise that bubble pushing is also related to the melt flow around the large bubble. As shown in Supplementary Movies 1 and 4,

small bubbles circulate around the large bubble due to Marangoni flow, indicating that the large bubble can be pushed in the melt pool with a downward force vector against the buoyancy force due to the high-shear flow between the bubble upper surface and the melt pool surface.

## Bubble migration mechanisms

We tracked the 2D projections of the bubble movements during the DED process from the radiographs, as shown in Fig. 4. As mentioned in the pore behaviour section, depending on the regions of the melt pool, bubbles are observed to migrate laterally or circularly. Based on this observation, we divided the melt pool into three regions, namely regions A (front), B (middle) and C (back), as shown in Fig. 4a. We then tracked the movements of a bubble which passed through these three regions as shown in Fig. 4b–d. In region A (Fig. 4b), which is the front of the melt pool, the bubble is observed to circulate counter clockwise, and the maximum velocity is measured to be ~88 mm s⁻¹, driven by Marangoni flow in the front of the melt pool.

The bubble then moves into region B (Fig. 4c), which is the middle of the melt pool; the bubble is observed to move up and down. We hypothesise that there are 4 flow cells: front, back, and one at each side. In the middle region, the bubble is in one of the side flow cells, and is going up/down and in/out of the page. In this region, the pore appears stationary in the laser frame of reference, which means it is pushed forward by the rear recirculation at the speed the substrate is moving (2 mm s⁻¹). At some stage, the drag at the bottom moves the pore back into the rear recirculation flow. This backward migration will be a balance of the recirculation flow (>10 mm s⁻¹), substrate motion (2 mm s⁻¹), and the capillary force originating from the thermal gradient.

When the bubble finally moves into region C, which is the back of the melt pool, its circular motion is observed to be clockwise, and its maximum velocity is ~196 mm s⁻¹, driven by the Marangoni flow in the rear of the melt pool, as shown in Fig. 4d. In Supplementary Movie 5, this small bubble coalesces with the large bubble, and the large bubble is formed by the coalescence of small bubbles.

We measured the instantaneous velocity of the bubble. We observed that the velocity of the bubble oscillates and accelerates between the highest and lowest points in each cycle, and decelerates when the bubble approaches these peaks. The mean and maximum velocity of the bubble in region B are measured to be 51 mm s⁻¹ and 145 mm s⁻¹, respectively. These values are higher than the corresponding velocities of 28 mm s⁻¹ and 88 mm s⁻¹ in region A.

## Bubble escape and entrapment mechanisms

Some bubbles follow the Marangoni-driven recirculating flow in the melt pool up to the surface and escape (Fig. 5a and see the video in Supplementary Movie 6). Some bubbles coalesce into large bubbles as discussed above, and some are entrapped into the solidification front.

Bubbles will escape if the buoyancy force is greater than the downward component of the recirculating cell. Another important factor, we hypothesise, is the location and velocity of the bubble inside the recirculation cell, as this also affects the upward component of the bubble, which ranges from 88 mm s⁻¹ (Fig. 4b) to 247 mm s⁻¹ (Fig. 5a) when the bubble changes from recirculation mode to escape. This indicates that the maximum bubble velocity in the vertical direction will affect bubble motion and hence escape.

In Fig. 5b, we compared the number of small bubbles that escape, coalesce, and are entrapped versus time, and the bubble versus time was defined as bubble rate. The number of bubbles for escaping and coalescing was observed to increase linearly with time. More bubbles escaped than were entrapped, as the bubble rate of 1.09 # ms⁻¹ in escaped bubbles is higher than the bubble rate of 0.05 # ms⁻¹ in entrapped bubbles. In Supplementary Fig. 7, the bubble rate by counting is 0.07 # ms⁻¹ in coalesced bubbles, indicating that more

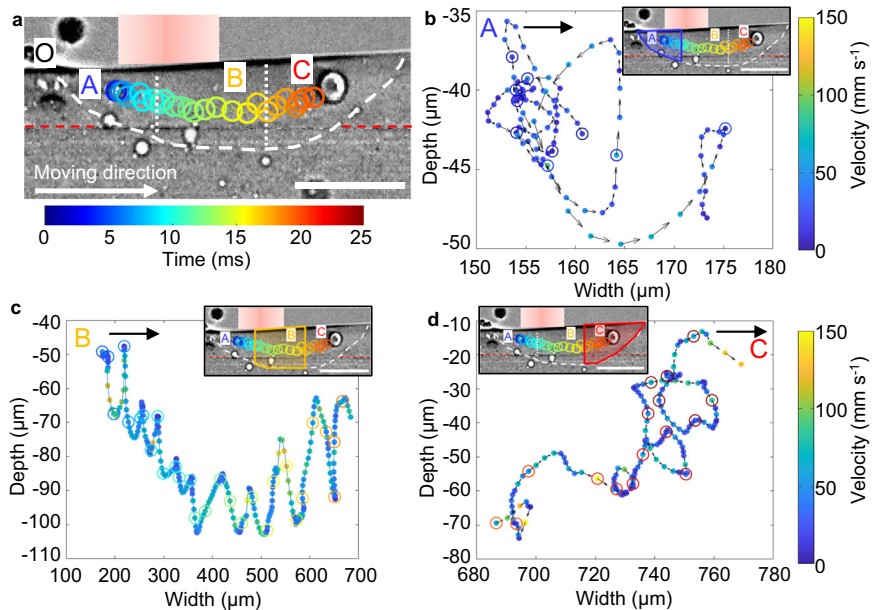

**Fig. 4 | Bubble migration from front to back of the melt pool. a** The melt pool is divided into regions A (front), B (middle) and C (back). The location of point O in the left intersection of the solid/liquid/air boundary was regarded as the starting position (depth = 0, Width = 0). Laser power is 160 W and traverse speed is 2 mm s⁻¹, layer 3. See the video in Supplementary Movie 5. **b** Motion track and velocity of the bubble circulation in region A, the velocity value is shown in the colour bar, and the arrow shows the moving direction. **c** Motion track and velocity of the bubble in region B. **d** Motion track and velocity of the bubble circulation in region C. The scale bars in (**a**) and inset figures in (**b**–**d**) are 300 μm.

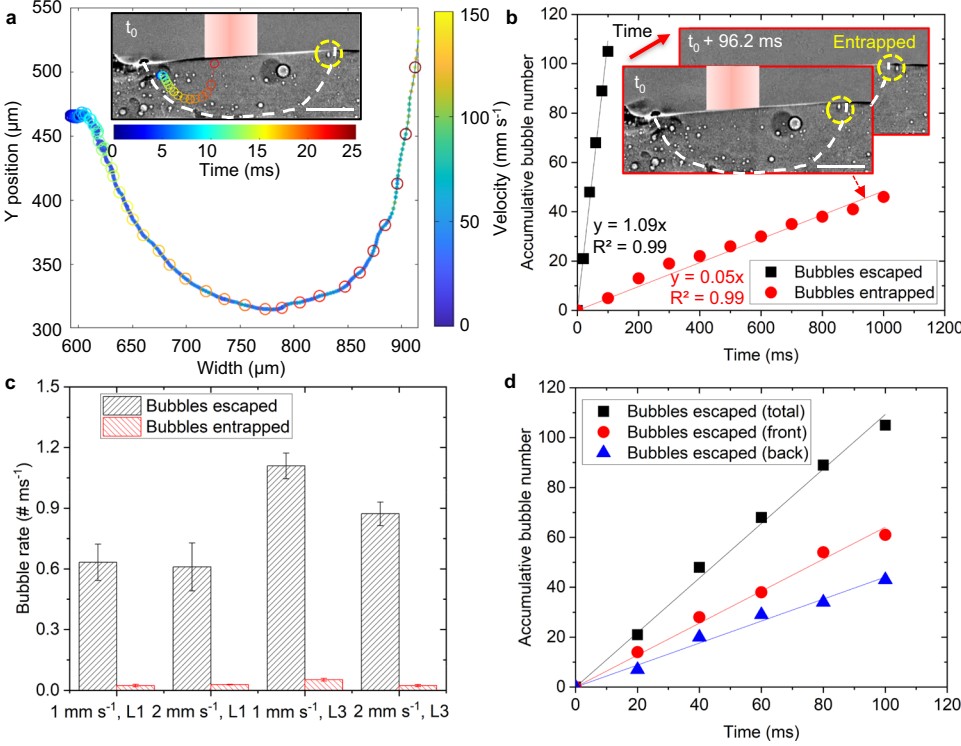

**Fig. 5 | Bubble escape from the melt pool and entrapment by the solidification front. a** Motion track and velocity of a bubble escape following Marangoni flow. The velocity value is shown in the parula colourmap. The time is shown in the jet colourmap. See the video in Supplementary Movie 6. **b** Accumulative number of bubble escape, coalesce and are entrapped with increasing time, and it is fitted linearly. Entrapped bubbles are shown in the inset figure. **c** The rates of bubble escaped and entrapped in a traverse speed of 1 mm s⁻¹, layer 1; 2 mm s⁻¹, layer 1; 1 mm s⁻¹, layer 3; 2 mm s⁻¹, layer 3, respectively. Error bars represent standard deviation. **d** Accumulative number of bubble escape in total, front and back of melt pool. In (**a**, **b**, **d**), the laser power is 160 W and the traverse speed is 1 mm s⁻¹, layer 3. Scale bars in (**a**, **b**) are 300 μm.

bubbles coalesced than were entrapped but less than escaped. The bubble number for coalescence by counting is in the range of the bubble number calculated using the large coalesced bubble volume divided by the initial bubble volume with the minimum, mean and maximum diameters of 17, 31 and 55 μm, and is close to the bubble number calculated by the mean diameter (Supplementary Fig. 7). This indicates that the counting results capture the average bubble coalescence behaviour under the conditions studied. We also compared the number of escaped bubbles per unit time against processing parameters, as shown in Fig. 5c. In layer 1, the rate of bubble escape and entrapment is shown to be constant despite the differences in traverse speed. The rate of both bubble escape and entrapment in the 3rd layer (L3) is higher than in the 1st layer (L1). We hypothesise that this is due to more bubbles being present in the tracks laid during the experiment than in the industrial machine-built substrate for layer 1. There is no significant difference in the bubble behaviour as a function of traverse speed.

We also investigated where the bubbles escape from the molten pool (Fig. 5d). More bubbles are observed to escape from the front of the melt pool. This could be due to the different velocities of the Marangoni flow in these two regions, and bubbles could stay longer at the back of the melt pool.

## Melt flow and bubble behaviour revealed by multiphysics modelling

The multiphysics model developed (based on ref. 50) uses a control volume solution of the mass, momentum and temperature transfer in the DED process, including phase change, bubble migration and coalescence, and powder particle impact on the surface of the molten pool. Full details of the model are in 'Methods' and Supplementary Information. We used this high-fidelity multiphysics model of DED[50] to validate the hypotheses we have formulated from the in situ X-ray imaging experiments on melt pool flow and bubble formation mechanisms.

## Melt pool recirculating flow cells

Figure 6a shows an X-ray radiograph of the melt pool, together with schematic arrows showing proposed Marangoni-driven recirculating flows at the front and back of the melt pool. Figure 6b shows a schematic illustration of our hypothesis above that there are four main recirculation flow cells in the melt pool, with two cells at the centre in and out of the page of the radiograph in Fig. 6a. The model predicted flows are shown in Fig. 6c, d, predicting recirculating flow cells at the front and back of the pool. These predictions match the X-ray results shown in Figs. 1 and 6a (also see videos in Supplementary Movies 1 and 2), where the pores recirculate in the front and back of the melt pool. The model also predicts two more flow cells, shown in a front view cut at the centre of the melt pool (Fig. 6e, f). This matches our hypothesis that there are two into and out of the page flow circulations, and explains the pores oscillation up and down in the middle zone in Fig. 4, as bubbles migrate from the front to the back of the melt pool.

As shown in Supplementary Fig. 8, the melt pool region was divided into the surface region and the inner periphery region. The temperature is higher in the surface region than in the inner periphery region. The flow velocity is higher in the region near the surface and generally increases with increasing temperature. The magnitude of the predictions of the flow also nicely matches the measured ones, as shown in Supplementary Fig. 8b, for a bubble with a diameter of 160 μm, where the average velocity is 100 mm s$^{-1}$ (20 - 400 mm s$^{-1}$), which is consistent with the velocity that we measured by X-ray imaging (Figs. 4 and 5).

## Bubble coalescence

Our hypotheses on bubble behaviour were also investigated with the model. One typical phenomenon is bubble coalescence. The bubble coalescence behaviour was investigated by first simulating the flow without bubbles to establish the four recirculating flows (see Fig. 6), and then bubbles were inserted at varying positions into the melt pool. Our observations of bubble coalescence were replicated in the model, showing that when 3 separate pores are placed in the flow, they are all driven towards the centre of a recirculation cell and coalesce (Fig. 7a, Supplementary Fig. 9 and Supplementary Movie 7). The front view shows this most clearly, with two bubbles coalescing to form a dumbbell shape. Due to surface tension, this shape is transient, quickly converting to a near-spherical large bubble.

For bubbles in the mid-front but the deep location (Supplementary Fig. 10 and Supplementary Movie 8), the bubbles are pushed between the front and side recirculation cell, where the flow velocity is lower, with a high flow velocity above. In this front-deep location, two bubbles also coalesce into a larger bubble, indicating this is conducive to coalescence. For bubbles in the middle location (Supplementary Fig. 11), bubbles are trapped in the centre of recirculation, and the local velocity is low, and bubble coalescence also occurs. These phenomena are similar to the bubble coalescence that occurred in the back of the melt pool. Therefore, bubble pushing occurs in back, front-deep and centre locations, as the Marangoni flow circulations can push bubbles down. Bubble coalescence is much more likely to occur in a larger melt pool of DED than in LPBF, as the residence time of bubbles is much greater, enabling them to coalesce to form large bubbles. The strong recirculating flow in a large pool constrains both the small and large bubbles' flow, creating conditions appropriate for bubble collision, with coalescence occurring when the film of liquid between colliding bubbles ruptures[60]. Coalescence reduces the overall free energy as it minimises the total bubble surface area[60].

## Bubble pushing at the surface

One surprising experimental observation was that large, coalesced bubbles did not immediately rise to the surface (due to buoyancy force) and pop. We hypothesised that this was due to the constriction of high-shear Marangoni flow. To test this a large bubble was put close to the surface in the back region, as shown in Fig. 7b, Supplementary Fig. 12 and Supplementary Movie 9. The model predicts that the shear flow circulates over the large bubble and pushes it in the melt pool. This pushing behaviour is consistent with the experimental results shown in Figs. 1 and 2, which confirms that the Marangoni flow contributes to pushing bubbles down in the melt pool, overcoming the buoyancy force, until the bubble reaches a critical size. Therefore, although bubble coalescence and growth can contribute to the larger buoyancy, the bubbles constrict the Marangoni flow, causing a downward force on the bubbles that delays their escape.

In Supplementary Discussion 2, the force balance onto the large bubble is calculated by comparing static buoyancy, shear and pressure forces induced by the molten metal flow. According to the corresponding simulation results of these forces in Supplementary Table 3, the large horizontal shear force can push the large bubble in the horizontally backward direction. The strong transverse Marangoni flow above the bubble pushes the bubble downward, balancing the buoyancy and positive shear force in the vertical direction. Therefore, the bubble can be pushed downward and backward when this flow structure is formed.

## Large bubble escape

Figure 7c (and Supplementary Fig. 13) shows an example where a very large bubble can escape from the top of the melt pool. The large bubble touches the top liquid surface when the bubble grows into a critical size and moves by the flow disruption, and then the top liquid surface ruptures to release the large bubble. Here the bubble is both very large (and hence large buoyancy force) and is located in the

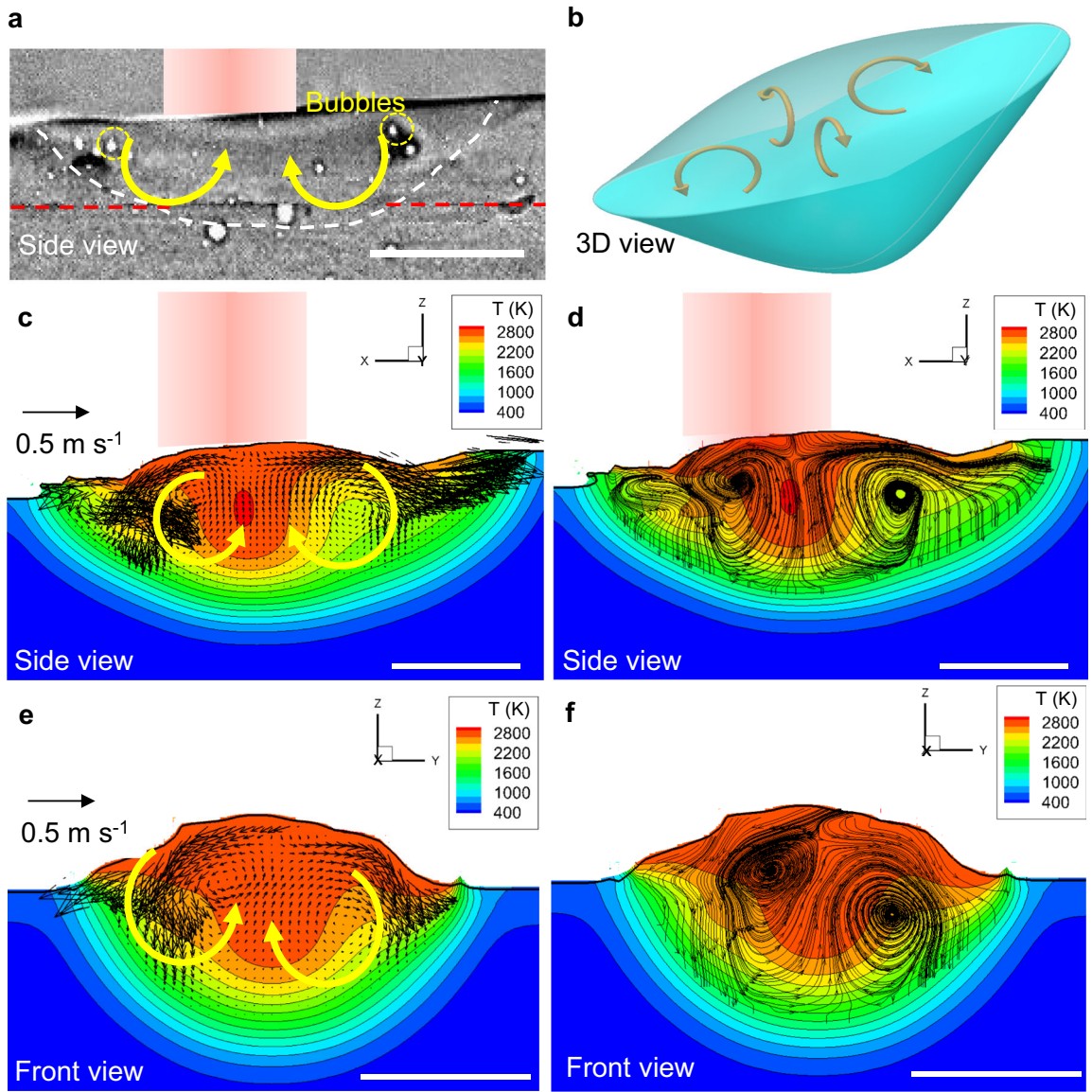

**Fig. 6 | Modelling results showing the melt pool flow without bubbles during DED. a** A X-ray image showing the melt pool. See the video in Supplementary Movie 2. **b** 3D view schematic showing the melt pool flow. **c** Side view and (**d**) corresponding 2D projected streamlines by modelling. **e** front view and (**f**) corresponding 2D projected streamlines by modelling. T in (**c–f**) represents temperature in K. The traverse speed is 2 mm s⁻¹, and the laser power is 160 W. Scale bars in (**a**) and (**c–f**) are 300 μm.

middle of the melt pool, between the flow recirculation cells, breaking the balance of forces, so the bubble pops up, explaining the experimentally observed behaviour. Computational fluid dynamics simulation in the Supplementary Information (e.g., see Supplementary Fig. 13), show how changes in the Marangoni-driven flow cells can create conditions entrapping bubbles within the flow cell, or pushing them to the melt pool surface, rupturing.

Most bubbles escape through the top liquid surface of the melt pool, it requires the high-speed X-ray imaging with a frame rate of 20 kHz to capture these phenomena (see videos in Supplementary Movies 2, 3, 5 and 6), as the X-ray imaging at a low frame rate of 1 kHz may miss a short escaping period due to the fast bubble escaping speed of 247 mm s⁻¹ in Fig. 5. A large bubble also escapes in the rear of the melt pool (see Supplementary Movie 2). The large bubble in the rear of the melt pool grows close to the top liquid surface of the melt pool, and the powder particle hits the melt pool and disrupts the Marangoni flow near the large bubble to break the force balance, so the large bubble can escape.

## Influence of powder particles hitting the melt pool surface

One possible explanation for the cyclic bubble migration in Fig. 4c and the circulating motion in Fig. 4d could be the disruption of the Marangoni flow when feedstock powder particles hit the surface, locally quenching the pool and altering the thermal, and hence surface tension gradient. Figure 8 shows the modelling results of direct particle bombardment on the surface, causing surface oscillation and local flow disruption. As the melt pool flow is disturbed by the bombardment, the bubble migrates similarly to the experimental observations.

In Fig. 8, to consider powder-hitting effects in our modelling, two approaches including forced and direct bombardment cases were applied. Based on the forced case (see the details in 'Methods'), Fig. 8b plots the temperature field, and smaller flow cells were observed in the melt pool. The corresponding velocity and trace of a bubble are shown in Fig. 8c. The up-down migration of a bubble under forced oscillation on the surface and migration from the front to the back of the melt pool, caused by the formation of circulation cells, which is consistent with experimental flow result that is shown in Fig. 8a, c. This indicated

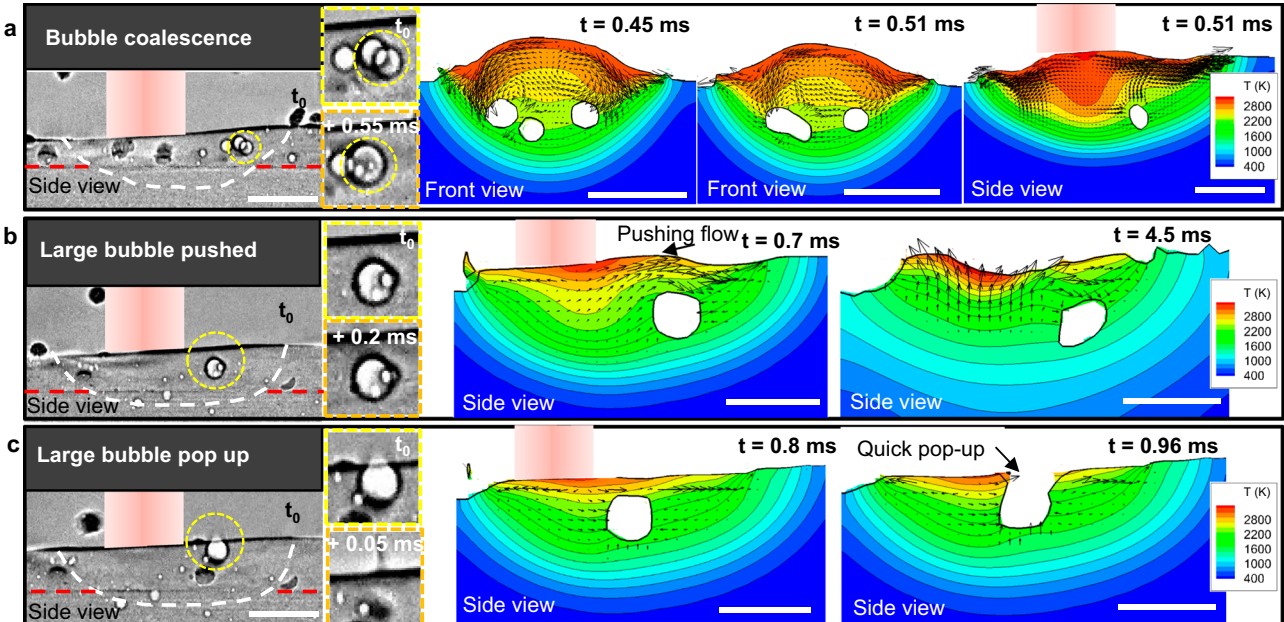

**Fig. 7 | Comparison between experimental data and modelling results of bubble coalescence, push-down and pop-up in the back of the melt pool.**
**a** X-ray images and corresponding simulation images showing bubble coalescence at $t = 0.45$ ms and $t = 0.51$ ms (shown in the front and side view images) (see simulation images in Supplementary Fig. 9 and video in Supplementary Movie 7). **b** X-ray images and corresponding simulation images showing a large bubble pushed by Marangoni shear flow at $t = 0.7$ ms and $t = 4.5$ ms from bubble insertion $t = 0$ ms (shown in the side view image) (see simulation images in Supplementary Fig. 12 and video in Supplementary Movie 9). **c** X-ray images and corresponding simulation images showing the large bubble pop-up at $t = 0.8$ ms and $t = 0.96$ ms from bubble insertion $t = 0$ ms (shown in the side view image) (see simulation images in Supplementary Fig. 13). T in the colour bar represents temperature in K. Scale bars in (**a–c**) are 300 μm.

that the phenomena can be attributed to the velocity and temperature perturbations induced by powder particles hitting.

For the direct bombardment case, as shown in Fig. 8d, e, the temperature field and flow direction near the powder change significantly. This can disrupt the normal Marangoni flow instantly and locally. As a result, the bubbles oscillate up and down and do not follow the normal circulating path. In addition, in the modelling results shown in Fig. 8e, an outward flow cell forms near hitting particles. In Fig. 8e, f, these flow cells can drive bubbles to migrate from the front to the rear of that melt pool (in the region indicated with a red dashed box) and then circulate outward (in the region indicated with a black dashed box). These phenomena are consistent with the experimental results of bubble migration in Figs. 8f and 4d. These results indicate that the flow cells generated by the particle impact can promote the bubble migration.

When the powder particle hits the melt pool, it can mainly generate two effects, namely, (1) the impact ripple waves of the particle when the powder particle just touches the melt pool surface and subsequent standing waves, which can affect the flow and bubble migration near the particle; (ii) after that, the powder particle gradually melts and quenches the melt pool, which can change the local temperature and flow pattern and bubble migrations near the particle. As shown in Figs. 8a–c and 4, the modelling results considering velocity and temperature perturbations for the powder effects are consistent with experimental results, in which the impact wave of powder particle causes the initial flow disruption and small flow cells (in accordance with the standing wave generation) are formed (Fig. 8d–f).

The motion trajectory in Figs. 4c and 8c is supposed to be mainly related to the simultaneous effects of Marangoni flow cells and powder impact effects. Although the powder particles can hit different locations of the melt pool at different times, the powder flow rate is controlled to be constant and high, which can produce a relatively consistent powder hitting, thus to change the flow pattern in the melt pool. It is also speculated from the experimental results that the later standing wave formation is nearly similar although the initial ripple

formation and the temperature effect occur in random places. Therefore, the bubble motion trajectory exhibits an organised pattern.

In summary, we have applied in situ high-speed synchrotron X-ray imaging and multi-physics modelling to reveal pore behaviours in the DED process, including pore formation, bubble coalescence and growth, pushing, migration, escape and entrapment. We found that the majority of bubbles in the melt pool originated from argon pores in the feedstock powder. Although many of these small bubbles escaped from the melt pool surface, some were entrapped by the solidification front and some coalesced into larger bubbles; those entrapped in the solid are often entrained in the pool in the next layer of track. The large bubbles are formed by up to one hundred small bubbles coalescing, and are pushed ahead of the solidification front until they reach a critical size. High-fidelity multi-physics modelling demonstrates that the constriction of the Marangoni shear flow between the melt pool surface and the top of the large bubbles provides sufficient downward force to overcome the upward buoyancy force, keeping the bubble entrained in the pool. After the bubble reaches a critical size, it interacts with the recirculating flow along the bottom of the melt pool, and is pushed to the pool surface and then pops out. We demonstrate the growth of large bubbles through coalescence and their subsequent periodic escape is a function of pool size and hence build conditions, including laser power and traverse speed. Although some prior studies of DED mention feedstock pores might be entrained, it is only through the in situ observations shown here that the key phenomena of bubble coalescence to form large pores have been revealed. This coalescence of up to 70 pores with a diameter of 20–50 μm to form a single 180 μm pore may control final component properties.

The bubble dynamics also includes their interaction with the fluid flow causing their entrainment or escape from the surface, and their interactions with solid/liquid interface, causing entrapment or pushing. To the best of the authors' knowledge, no bubble coalescence and growth in a large melt pool of AM was reported in previous studies. The solid/liquid interface entrapment or pushing of bubbles was

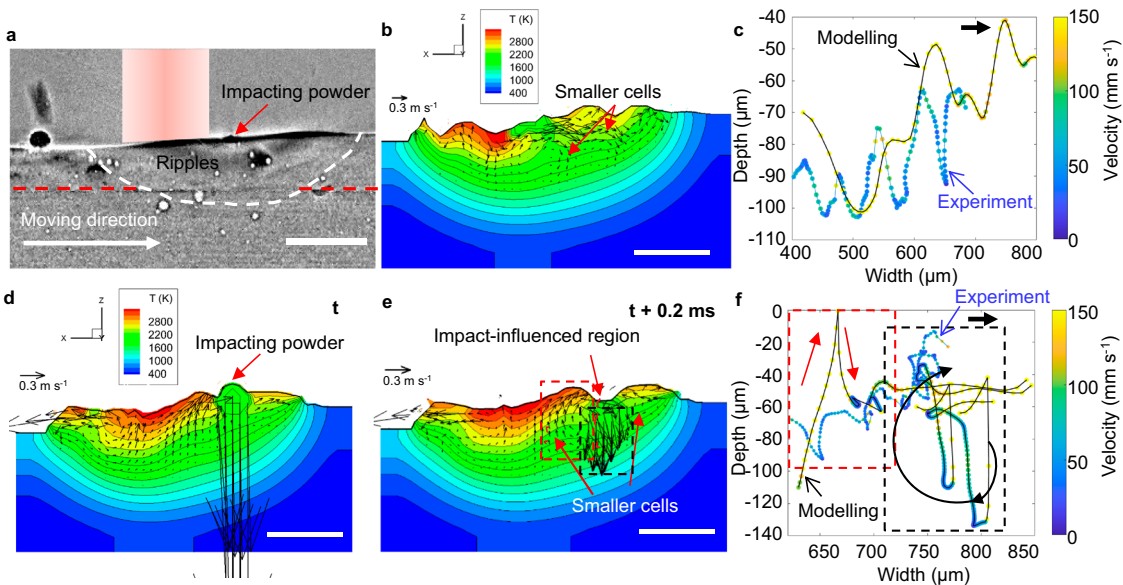

**Fig. 8 | Comparison between experimental data and modelling results of bubble migration in the melt pool. a** A radiograph showing melt pool with impacting powder. **b** For the forced case, the temperature field obtained by modelling with the same parameters as the X-ray imaging experiments, velocity and temperature perturbations given to the surface to simulate powder hitting effects, and (**c**) corresponding velocity and trace of a bubble inside the melt pool. The up-down migration of a bubble under forced oscillation on the surface, caused by the formation of circulation cells compared with the large Marangoni circulation shown in Fig. 6c, d. Modelling and experiment results are shown in blue and black lines, respectively. **d** Temperature field considering impacting powder at *t*, (**e**) formation of smaller cells at *t* + 0.2 ms. And (**f**) corresponding velocity and trace of a bubble inside the melt pool. The modelling and experimental curves are connected in black and blue lines, respectively. Direct simulation of random powder bombardment where sudden velocity increase is induced in the impact region, which causes irregular bubble migration such as the up-down migration and local circulation. Modelling and experiment results are shown in blue and black lines, respectively. For the forced case, the (circular) surface wave period is set as 0.6 ms, surface wave number is 5 in the pool lateral direction. For the direct bombardment case, the impacting velocity is 4 m s⁻¹, the powder diameter is 90 μm, the impacting interval is 0.5 ms and the powder temperature is 1800 K for simplicity. These values for modelling are determined by the X-ray imaging experimental video. T in the colour bar in (**b**) and (**d**) represents temperature in K. The velocity unit in (**b**, **d**, **e**) is m s⁻¹. Scale bars in (**a**, **b**, **d**, **e**) are 300 μm.

reported in directional solidification[13,25], but direct observation has not been reported in DED. Bubble entrainment, escape and entrapment in the solid were seen for keyhole pores in LPBF[33], but not in DED.

The bubble behaviour should be related to the Marangoni flow in the melt pool. The Marangoni flow was observed by Mills et al.[58] and Lee et al.[61] using ex situ observations, and modelled by Paul and Debroy[62], and more recently in situ observations by Aucott et al.[63] for welding and Guo et al.[64] in LPBF. However, our observations in DED also elucidate that some small bubbles follow the flow, some float out, some are entrapped, and some coalescence; whilst the large bubbles stay in the melt pool. This study contributes to a greater fundamental understanding of pore evolution and dynamics mechanisms during additive manufacturing processes, providing a potential pathway for developing a pore minimisation strategy for the DED process.

## Methods

### Material characterisation
The gas atomised nickel-based superalloy RR1000 powder was characterised with scanning electron microscopy (SEM) JEOL JSM-6610V. The SEM image of the powders and corresponding powder size distribution was plotted in Supplementary Fig. 2. The powders were segmented using Otsu's method and then separated using a watershed algorithm in MATLAB to measure the powder size.

### Blown Powder Additive Manufacturing Process Replicator II (BAMPR II) system and processing conditions
In situ synchrotron X-ray imaging was performed on the ID19 beamline at the European Synchrotron Radiation Facility (ESRF) to capture the pore dynamics and formation during DED. BAMPR II was a custom-designed system to replicate the commercial DED process that can be integrated into synchrotron beamtime. It includes an environmental chamber (Saffron, Scientific Equipment Ltd), a high-precision 3-axis platform (Aerotech, US), a coaxial DED nozzle, and a Ytterbium-doped laser (SPI lasers Ltd, UK) in continuous wave mode with a wavelength of 1070 nm and a maximum power of 200 W. The beam reducer (Optogama, Lithuania) was equipped to focus the beam size down to 400 μm with a symmetric Gaussian shape. The laser beam spot size is defined with 1/e², and the profiled laser beam is plotted in Supplementary Fig. 14. The measured beam spot size is about 390 μm near the focal point. The environmental chamber was filled with argon gas to reduce oxidation, and the oxygen level was generally controlled to be below 10 ppm during the experiments. The powder was delivered to the nozzle in a stream of argon gas by the industrial powder feeder system (Oerlikon Metco TWIN-10-C) and then blown to normal to the substrate plate. The powder feed rate in this work is 1.8–2.7 g min⁻¹. The substrate with dimensions of 60 mm × 20 mm × 1.5 mm was mounted in a moving platform with a maximum traverse speed of 50 mm s⁻¹. The high-speed imaging for the melt pool and pores was captured at spatial (4 μm) and temporal resolutions (20 kHz) using a CMOS camera (type: SAZ, Photron, Japan) lens-coupled to a LuAG:Ce single-crystal scintillator. The low-speed imaging was captured at spatial (3.7 μm) and temporal resolutions (1 kHz) using a CMOS camera (type: Dimax, PCO AG, Germany) lens-coupled to a LuAG:Ce single-crystal scintillator as well to observe a longer duration period.

### Image processing
The acquired radiographs were processed using ImageJ and MATLAB. A flat field correction was conducted via the equation: $FFC = (I_0 - Flat_{avg}) / (Flat_{avg} - Dark_{avg})$, where $FFC$ is the flat field corrected image, $I_0$ is the raw image, $Flat_{avg}$ is the average of 100 flat field images (imping beam profile without sample) and $Dark_{avg}$ is the

average of 100 dark field images (sensor noise without any impinging radiation).

## Multi-physics modelling

The temperature, velocity and bubbles in the melt pool were simulated using multi-physics modelling which is validated with experimental parameters[50]. The fluid flow equations of mass, momentum and temperature are solved along with interface capturing by the Coupled Level-Set/Volume-Of-Fluid (CLSVOF) method:

$$(\text{mass})\frac{\partial \rho}{\partial t} + (\boldsymbol{u} \cdot \nabla)\rho = -\rho\nabla \cdot \boldsymbol{u} \qquad (1)$$

$$(\text{momentum})\frac{\partial \boldsymbol{u}}{\partial t} + (\boldsymbol{u} \cdot \nabla)\boldsymbol{u} = -\frac{\nabla p}{\rho} + \boldsymbol{Q_u} + \boldsymbol{g} + \boldsymbol{F_{u,\text{surf}}} \qquad (2)$$

$$(\text{temperature})\frac{\partial T}{\partial t} + (\boldsymbol{u} \cdot \nabla)T = -\frac{p\nabla \cdot \boldsymbol{u}}{\rho c_p} + Q_T \qquad (3)$$

where $\rho$ is the density, $\boldsymbol{u}$ is the velocity, $T$ is the temperature, $p$ is the pressure and $c_p$ is the constant-pressure heat capacity. $\boldsymbol{Q_u}$ represents the Newtonian viscous force and Darcy's force in the mushy zone, $\boldsymbol{g}$ is the gravitational acceleration and $\boldsymbol{F_{u,\text{surf}}}$ represents the interfacial surface tension force including the Marangoni effect. $Q_T$ represents the heat transport, including heat conduction by Fourier's law, viscous work, latent heat for phase change and radiation. The laser power is given to the melt pool surface by the ray tracing method. Material accumulation on the surface is calculated by the conservation of mass. Details of the numerical method can be found in Supplementary Information and ref. 50. The physical properties such as viscosity and thermal conductivity are derived as in ref. 65. The fine grid resolution is 16 µm. The resolution for long-time simulation is 32 µm in cases of Figs. 7b, c and 8b, d, e. Still, we justify the use of this grid since we have confirmed that the same Marangoni flow structure can be reproduced. For the small bubble tracking cases in Fig. 8b, d, e, these bubbles are assumed to be sufficiently small that they can be treated as Lagrangian point particles (see Supplementary Method 1 and Supplementary Fig. 15 for justification). For the large bubbles (e.g., those in Fig. 7), the bubbles are explicitly modelled using the level-set method to capture the liquid gas interface, simulating the surface shape and bubble coalescence (see Supplementary Method 1).

In the forced case in Fig. 8, velocity and temperature perturbations were directly applied to the melt pool surface. From the experimental observation, standing waves are seen after particle bombardment. For simplicity, the perturbations to give on the surface are set as follows; the wavelength $\lambda$ is one-fifth of the longitudinal melt pool length (5 standing waves in the melt pool), the period $T$ is 0.6 ms, the displacement amplitude $A$ is 30 µm, and the velocity amplitude is $A\omega$, where $\omega = 2\pi/T$. In the assumed region of particle bombardment, the surface temperature is set at 1800 K, but this temperature effect is minor.

## Data availability

The authors declare that the data supporting the findings of this study are available within this article and its Supplementary Information file, or from the corresponding authors upon request.

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

## Acknowledgements
This research is financially supported by the Engineering and Physical Sciences Research Council (EPSRC) via MAPP: Future Manufacturing Hub in Manufacture using Advanced Powder Processes (EP/P006566/1), Rolls-Royce plc. via the Aerospace Technology Institute program REIN-STATE (contract 51689), and the Royal Academy of Engineering (CiET1819/10). We also acknowledge the use of facilities and support provided by the Research Complex at Harwell and thank the ESRF for providing the beamtime proposal (MA-4857) and the staff at ID19 beamline for technical assistance. This work is partially supported by Next Generation TATARA Project sponsored by the Government of Japan and Shimane Prefecture. C.P. would like to acknowledge the funding from Innovation Fellowship funded by Engineering and Physical Science Research Council (EPSRC), UK Research and Innovation, under the grant number: EP/S000828/2.

## Author contributions
K.Z., M.A.J. and P.D.L. conceived the research. K.Z. wrote the manuscript. K.Z. and P.D.L. finalised the manuscript, with all authors contributing. K.Z. led and performed data analysis and image processing (with help from P.D.L., C.L.A.L., Y.C., X.F. and S.B.). The experiments were performed remotely during the lockdown, with all authors virtually participating, but S.M., Y.C., M.F., M.M., B.L., K.J. and A.R. who were based at ESRF physically present. J.S. and C.P. performed modelling.

## Competing interests
The authors declare no competing interests.
