## [Peer Review File · Nature Communications]

Pore evolution mechanisms during directed energy deposition additive manufacturingREVIEWER COMMENTS

Reviewer #1 (Remarks to the Author):

Overall this is a comprehensive study with in-situ X-ray imaging, multiphysics modeling and very detailed analysis (particularly some quantitative analysis). Below are some comments for consideration.

There have already been many reports on pore evolution in metal AM by in-situ observation experiments and multiphysics modeling, mostly on LPBF and some on DED. Most of the pore evolution mechanisms, particularly the driving forces and motions of bubbles in the molten pool, are similar between LPBF and DED, and the differences mainly include the absence of keyhole (conduction mode is simpler in physics), larger molten pool, and powder bombardment in DED, which are the new contributions of this work.

My opinions on the novelty of observation and understanding into the physical mechanisms are as follows:

- 1) The pore formation from the powder is well-known and not surprising.
- 2) Bubble coalescence and growth is uncommon in previous literatures of AM, mainly because that the molten pool was not big enough for bubbles to coalesce.
- 3) Solid/liquid interface pushing of bubbles was observed in directional solidification, and this may be the first-time observation in AM, mainly because that the molten pool was not big enough to achieve big pores in previous studies.
- 4 & 5) Bubble entrainment, escape, and entrapment (in the solid) have also been well understood in previous literatures.

The most critical issue is to justify that the experimental setup is close to the industrial applications. While the authors are not wrong to criticize that the previous works on high-speed X-ray imaging of DED process is not representative, how well can this work mimic industrial cases? The substrate traverse speed of 1 mm/s or 2 mm/s is lower than those commonly used speed range of 6-20 mm/s, the laser spot size of 400 μm is also much smaller than normal values of 1-3 mm, and the laser power of 160 W is much lower than the common values of 500-3000 W. Moreover, another important parameters, the powder feed rate (it is 1.8-2.7 g/min in supplementary document) and powder velocity, are not given. Even if the energy density (depending on the definition) may be similar to those of industrial cases to get the conduction-mode molten pool, the molten pool is not as large, and the powder bombardment is not as fierce, as in industrial applications. These two issues are significant differences between DED and LPBF. Especially if the powder feed rate is high and highly concentrated in the center, the powder impact may significantly alter the molten pool flow and also entrain gas into the molten pool, thereby changing the bubble evolution.

As X-ray imaging is 2D projection, overlap of bubbles in the projection may not necessarily be coalescence, while the growth is more solid evidence.

The detail of the model is not sufficient. How were the velocity and temperature perturbations determined and added in the forced case?

In Fig.8 b and e, the bubbles cannot be seen in the simulation, while the bubble in fig. 8d is not as big as that observed in Fig. 8a.

How well (in terms of the bubble sizes) was the solid/liquid interface trapping/pushing of bubbles reproduced in the simulation?

Reviewer #2 (Remarks to the Author):

In this paper, high-speed synchrotron x-ray imaging and multi-physics modeling were used to

characterize the pore evolution mechanisms during the directed energy deposition process. The results are interesting and shed light on the pore minimization strategy development. Here are some comments for the authors to consider.

1. The travel speed of the substrate in Fig. 1 is 2 mm/s. I did a quick calculation and the laser-matter interaction time is 0.2 second. It may indicate a nearly symmetrical melt pool geometry with respect to the laser beam. It is suggested to provide the laser beam spot size definition, e.g., $1/e$, $1/e^2$, and confirm the laser beam locations (red color) in related X-ray images and schematics.
2. In Fig. 2e, the lower travel speed of the substrate (1 mm/s) shows a much higher bubble formation rate from previous layers. It is largely attributed by the authors to the larger melt pool size. Related data like melt pool sizes could be provided.
3. Figs. 2e and 5c show an interesting trend in bubble number reduction. What would happen if the laser beam scans the same track without powder feeding for a second time or for many times? Will this help finally eliminate the bubbles?

Reviewer #3 (Remarks to the Author):

This work studies the pore evolution behavior during the directed energy deposition (DED) additive manufacturing process through a combination of in-situ X-ray imaging and multi-physics simulation. The pore formation and evolution during DED have been studied by in-situ X-ray imaging before. A representative work by S. Wolff et al. (International Journal of Machine Tools and Manufacture, volume 166, 2021, 103743, <https://doi.org/10.1016/j.ijmachtools.2021.103743>) using in-situ X-ray imaging with enhanced spatial and temporal resolution at the Argonne National Lab provided a detailed analysis of pore formation and evolution mechanisms during the DED process. Although Wolff's work used a small laser beam (which is more suitable for LPBF process instead of DED) to produce a keyhole mode melt pool, I don't see why some of their findings could not be applied to a conduction mode melt pool. For example, regardless of the melting mode the melt pool is experiencing, the pore formation from the "porosity inside the feedstock powder" should all be valid and expected.

The pore evolution behaviors, as frequently stated in this paper, are strongly related to the Marangoni flow within the melt pool. However, the flow pattern in the conduction mode melt pool has been well studied before. For instance, the front and back outward recirculating flow cells were reported by L. Aucott et al. (Nature Communications, volume 9, 2018, 5414, <https://doi.org/10.1038/s41467-018-07900-9>), also using in-situ X-ray imaging. Similarly, Q. Guo et al. (Additive Manufacturing, volume 31, 2020, 100939, <https://doi.org/10.1016/j.addma.2019.100939>) revealed the side flows in conduction mode melt pool via in-situ X-ray imaging. With these groundworks, it is unsurprising that the majority of bubbles observed in this work should move in accordance with the flow patterns in the melt pool.

The paper also lacks in-depth analysis. For instance, in the section "Bubble migration mechanisms," the content is merely a description of experimental observations and measurements, lacking analytical insight. If there is no analysis, then what is described here should not be referred to as "mechanisms", but merely "behaviors". The same concern applies to the section on "Bubble coalescence." In the section "Bubble pushing at the surface", why the quantified numbers of the shear force and buoyancy force were not provided? Why was the hydraulic pressure not considered in the calculation? How does the shear force offset the buoyancy force, given that the shear force acts in the tangential direction of the bubble surface? In the section "Large bubble escape", further force analysis is required to support the explanation provided in Line 490: "breaking the balance of forces." Moreover, large bubble escape is not limited to this one circumstance. For example, in Supplementary Video 1, some large bubbles disappeared suddenly inside the melt pool, without touching the top liquid surface; in Supplementary Video 2, some escaping bubbles were not in the middle of the melt pool, but in the rear circulation. None of them are explained in the paper. In the section "Influence of powder particles hitting the melt

pool surface", what is the relative strength of impact wave versus temperature perturbations on the influence of bubble migrations? Which one is dominating, and under which circumstances? In addition, particle-hitting events are supposed to occur randomly in terms of impact location and timing. Why could such a random phenomenon possibly cause a near-regular bubble movement in Figure 4c with a nearly fixed frequency and wavelength of the motion trajectory?

With the above questions arising, I am not convinced that the work reported here has been properly interpreted, taking all details and scenarios into consideration.

The writing in the paper lacks organization and can be challenging to follow. For example, lines 145-153 are not relevant to pore formation and should be relocated. Descriptions in lines 382-385, 487-491 are overly vague and do not convey valuable information. Lines 525-528 are difficult to comprehend and could benefit from clarification or rephrasing.

Detailed Response to Reviewers' Comments

Ref. No.: NCOMMS-23-16900

Title: Pore evolution mechanisms during directed energy deposition additive manufacturing

Dear Reviewers:

The authors appreciate the time and effort that you dedicated to review this work. Many thanks for providing us with valuable comments, which helped us to improve this manuscript significantly. The point-by-point response to these comments along with substantial revisions made are listed below. Our responses are noted in blue text. The revised parts in this manuscript are **highlighted in yellow and also shown below the responses.**

Reviewer #1 (Remarks to the Author):

1. Overall this is a comprehensive study with in-situ X-ray imaging, multiphysics modeling and very detailed analysis (particularly some quantitative analysis). Below are some comments for consideration.

The authors appreciate the reviewer's positive and constructive comments for revision. All comments have been carefully addressed.

2. There have already been many reports on pore evolution in metal AM by in-situ observation experiments and multiphysics modeling, mostly on LPBF and some on DED. Most of the pore evolution mechanisms, particularly the driving forces and motions of bubbles in the molten pool, are similar between LPBF and DED, and the differences mainly include the absence of keyhole (conduction mode is simpler in physics), larger molten pool, and powder bombardment in DED, which are the new contributions of this work.

We appreciate the reviewer pointing out the main contributions to new understandings of DED in our work. To highlight the new findings, the introduction of this manuscript is enhanced. Please see the revisions below or on page 3 in the revised manuscript. These key novelties were not as clearly understood by Reviewer #3, therefore we have listed and flagged these in the introduction more clearly.

[Lines 61 - 68]: "Two studies systematically investigated pore formation during LPBF using high-speed X-ray imaging^{30,33}. It was found that pore formation can be caused by a critical instability at the bottom of the keyhole³⁰. However, this mechanism does not apply to the DED process which has a larger laser spot size and a lower energy density than LPBF. Hence DED is normally in conduction mode with no keyhole^{2,39}, has a much larger molten pool^{40,41}, and includes powder bombardment⁴² which can contribute to different bubble evolution and melt pool dynamics."

3. My opinions on the novelty of observation and understanding into the physical mechanisms are as follows:

1) The pore formation from the powder is well-known and not surprising.

2) Bubble coalescence and growth is uncommon in previous literatures of AM, mainly because that the molten pool was not big enough for bubbles to coalesce.

3) Solid/liquid interface pushing of bubbles was observed in directional solidification, and this may be the first-time observation in AM, mainly because that the molten pool was not big enough to achieve big pores in previous studies.

4 & 5) Bubble entrainment, escape, and entrapment (in the solid) have also been well understood in previous literatures.

We appreciate the reviewer listing the novelties of our observations, flagging where we use our *in situ* observations to validate previously hypothesised mechanisms, and where we've uncovered new ones. We have enhanced our manuscript to highlight the novelty by comparing it with previous studies, explained below for each point.

1) The pore formation from the powder is well-known and not surprising.

We agree it was known that the feed powder can have small argon pores, potentially causing porosity in the scan track; however, the only prior *in situ* study in DED that we know of concludes that pore transfer from the porosity inside powder is <0.5% of the total porosity¹. Our results show that for more industrial conditions pores from the powder are the major source. To clarify this, the corresponding revision was added on page 13 or below.

[Lines 245 - 251]: “For their conditions, using plasma atomised powders and laser conditions creating a keyhole, they concluded that feedstock porosity is a relatively insignificant contribution to the process with a contribution ratio of 0.22%⁵. Our results show that for the more industrial conditions used here, feedstock porosity becomes the major source of pores rather than a negligible one. This would be the one of major differences in pore formation between this work and the prior study⁵.”

2) Bubble coalescence and growth is uncommon in previous literatures of AM, mainly because that the molten pool was not big enough for bubbles to coalesce.

We agree. We are not aware of any prior mentions of bubble coalescence in AM literature, and we also agree this is probably due to the small melt pool size of prior studies. Yet this coalescence of up to 70 pores with a diameter of 20 - 50 μm to form a single 180 μm pore will control many final component properties and is novel and critically important to the field. We have added the paragraph below to the discussion on page 34.

[Lines 636 - 640]: “Although some prior studies of DED mention feedstock pores might be entrained, it is only through the *in situ* observations shown here that the key phenomena of bubble coalescence to form large pores have been revealed. This coalescence of up to 70 pores with a diameter of 20 - 50 μm to form a single 180 μm pore may control final component properties.”

3) Solid/liquid interface pushing of bubbles was observed in directional solidification, and this may be the first-time observation in AM, mainly because that the molten pool was not big enough to achieve big pores in previous studies.

We agree. This novelty has been highlighted in the discussion on page 35 or below.

[Lines 645 - 646]: “The solid/liquid interface entrapment or pushing of bubbles was reported in directional solidification^{13,25}, but direct observation has not been reported in DED.”

4 & 5) Bubble entrainment, escape, and entrapment (in the solid) have also been well understood in previous literatures.

We agree these phenomena are well known for keyhole pores in LPBF, but these are the first observations in DED, both for the small pores and we know of no prior studies or even hypotheses for the mechanisms for large-coalesced pores.

Please see the revisions below or on page 35 of the revised manuscript.

[Lines 641 - 648]: “The bubble dynamics also includes their interaction with the fluid flow causing their entrainment or escape from the surface, and their interactions with solid/liquid interface, causing entrapment or pushing. To the best of the authors’ knowledge, no bubble coalescence and growth in a large melt pool of AM was reported in previous studies. The solid/liquid interface entrapment or pushing of bubbles was reported in directional solidification^{13,25}, but direct observation has not been reported in DED. Bubble entrainment, escape and entrapment in the solid were seen for keyhole pores in LPBF³³, but not in DED.”

4. The most critical issue is to justify that the experimental setup is close to the industrial applications. While the authors are not wrong to criticize that the previous works on high-speed X-ray imaging of DED process is not representative, how well can this work mimic industrial cases? The substrate traverse speed of 1 mm/s or 2 mm/s is lower than those commonly used speed range of 6-20 mm/s, the laser spot size of 400 μm is also much smaller than normal values of 1-3 mm, and the laser power of 160 W is much lower than the common values of 500-3000 W. Moreover, another important parameters, the powder feed rate (it is 1.8-2.7 g/min in supplementary document) and powder velocity, are not given. Even if the energy density (depending on the definition) may be similar to those of industrial cases to get the conduction-mode molten pool, the molten pool is not as large, and the powder bombardment is not as fierce, as in industrial applications. These two issues are significant differences

between DED and LPBF. Especially if the powder feed rate is high and highly concentrated in the center, the powder impact may significantly alter the molten pool flow and also entrain gas into the molten pool, thereby changing the bubble evolution.

Many thanks for your suggestion, we have added a table listing the range of conditions we used, those used in industrial machines, and those in other published *in situ* rig setup. These show that our melt pool size, powder feed rate and other key parameters are of a similar scale to the low-mid industrial range, as is our Linear Energy Density (LED). Our X-ray imaging results in DED also show the conduction mode to match the industrial DED. In terms of the other *in situ* study of porosity (Wolff et al. ¹), the melt pool volume is nearly an order of magnitude larger, and no keyhole is observed in our work. Further, we image multi-track builds, which enabled mechanisms such as the entrapment of small pores, and subsequent remelting raised by reviewer #2 as well. It is challenging to cover the full industrial range in a single study, as higher laser power requires a thicker substrate, hence higher X-ray flux, which is limited by the synchrotron X-ray beam. A discussion of the comparison and contrast of these insights has been added to the text. Please see them on page 14, in Supplementary Discussion 1 on pages 19 and 22, Methods on page 36 or below. In our study, we also consider the powder impact effects on the melt pool flow and the bubble evolution in the powder particle hitting section which have been further enhanced on pages 30-33. We also agree with the reviewer's comments on the influence of powder impact, especially if focussed, on molten pool flow. The powder impact is discussed, although we did not study changes in the powder distribution (e.g., focussed in centre as suggested). For example, the main changes to the text in [Lines 572 - 599] are added below.

[Lines 268 - 270]: "Further discussions about the comparison between our work with the previous work ⁵ and the industrial DED can be found in Supplementary Discussion 1".

[Lines 175 - 189] in Supplementary information:

"Supplementary Discussion 1

As shown in Supplementary Table 2, the melt pool length (500 - 2000 μm) and depth (300 - 1200 μm) cover the most range of the industrial DED (200 - 3500 μm) and expand a larger range than Wolff et al.'s study ³. Our DED replicator can achieve multiple layers, traverse speed from 0.5 to 50 mm/s, and laser power from 0 to 500 W, powder feed rate from 1 to 5 g/min, which are also close to the industrial scales. The powder velocity from 2 to 10 m/s in our DED study is also in the industrial range. Our X-ray imaging results also show the conduction mode to match the industrial DED rather than the keyhole mode in the previous work ³. The energy density of the DED process in this work is nearly 125 - 500 J/mm² with an available range of 0 - 1250 J/mm², which covers the range of energy density of industrial DED processes with 4 - 189 J/mm², and also shows a larger range than Wolff et al.'s study ³. This demonstrates that the mechanisms observed in this work can be applied to the industrial-scale DED. The laser spot size of 400 μm is also close to the industrial scale of 400 - 11000 μm . One reason for choosing these parameters is also to achieve the optimal DED build and X-ray imaging observation quality to compromise the synchrotron X-ray beam."

[Lines 238 - 241] in Supplementary information:

"Supplementary Table 2. Comparison of key DED parameters between our *in situ* DED study with Wolff et al. ³ and industrial DED

	This in situ DED study	Wolff et al.'s study [³]	Industrial DED
Melt pool length (μm)	500 - 2000	950	200 - 3500 [⁴⁻⁶]
Melt pool depth (μm)	300 - 1200	280	200 - 2000 [⁴⁻⁶]
Track, layer #	1 - 3 (1 - n) *	1	n [⁷⁻¹⁴]
Traverse speed (mm/s)	1 - 2 (0.5 - 50) *	100	5 - 28 [⁷⁻¹⁴]
Laser power (W)	100 - 200 (0 - 500) *	156, 208	270 - 5000 [⁷⁻¹⁴]
Laser spot size (μm)	400 (100 - 700) *	100	400 - 11000 [⁷⁻¹⁴]

Powder feed rate (g/min)	1.8 - 2.7 (1 - 5) *	0.6	1 - 30 [7-13]
Linear energy density (J/mm²)	125 - 500 (0 - 1250) *	20.8	4 - 189 [7-14]
Powder shape	Spherical	Irregular	Spherical preferred [12,13,15,16]
Powder particle size (µm)	53 - 106 (30 - 150) *	75 - 200	30 - 150 [12,13,15,16]
Powder velocity (m/s)	2 - 10	1.5 - 3.2	2 - 30 [17-19]

* Values before the bracket are those used for the runs in this study. Values in brackets are the range available with BAMPR II."

[Line 681]: "The powder feed rate in this work is 1.8 - 2.7 g/min."

[Lines 572 - 599]: "For the direct bombardment case, as shown in Fig. 8d and e, the temperature field and flow direction near the powder change significantly. This can disrupt the normal Marangoni flow instantly and locally. As a result, the bubbles oscillate up and down and do not follow the normal circulating path. In addition, in the modelling results shown in Fig. 8e, an outward flow cell forms near hitting particles. In Fig. 8e and f, these flow cells can drive bubbles to migrate from the front to the rear of that melt pool (in the region indicated with a red dashed box) and then circulate outward (in the region indicated with a black dashed box). These phenomena are consistent with the experimental results of bubble migration in Fig. 8f and Fig. 4d. These results indicate that the flow cells generated by the particle impact can promote the bubble migration.

When the powder particle hits the melt pool, it can mainly generate two effects, namely, i. the impact ripple waves of the particle when the powder particle just touches the melt pool surface and subsequent standing waves, which can affect the flow and bubble migration near the particle; ii. after that, the powder particle gradually melts and quenches the melt pool, which can change the local temperature and flow pattern and bubble migrations near the particle. As shown in Fig. 8a-c and Fig. 4, the modelling results considering velocity and temperature perturbations for the powder effects are consistent with experimental results, in which the impact wave of powder particle causes the initial flow disruption and small flow cells (in accordance with the standing wave generation) are formed (Fig. 8d-f).

The motion trajectory in Fig. 4c and Fig. 8c is supposed to be mainly related to the simultaneous effects of Marangoni flow cells and powder impact effects. Although the powder particles can hit different locations of the melt pool at different times, the powder flow rate is controlled to be constant and high, which can produce a relatively consistent powder hitting, thus to change the flow pattern in the melt pool. It is also speculated from the experimental results that the later standing wave formation is nearly similar although the initial ripple formation and the temperature effect occur in random places. Therefore, the bubble motion trajectory exhibits an organised pattern."

5. As X-ray imaging is 2D projection, overlap of bubbles in the projection may not necessarily be coalescence, while the growth is more solid evidence.

The reviewer is correct that we are integrating through thickness and hence see overlapping bubbles, and that pore growth is the more definitive evidence. However, in the ESRF X-ray imaging data the frame rate of 20 kHz is sufficiently fast to see the small bubbles touch the larger pore and disappear, providing direct evidence of coalescence. The following text was added. Please see the revisions below or on page 16.

[Lines 304 - 308]: "The growth of the large pore provides convincing evidence of bubble coalescence, and although there may be some overlap of the bubble through the thickness, the high frame rate data shows small bubbles touch the larger pore and disappear, also providing strong evidence of coalescence, as shown in Supplementary Movie 3 at about $t = 41$ and 48 ms."

6. The detail of the model is not sufficient. How were the velocity and temperature perturbations determined and added in the forced case?

Although the model has been previously published in detail, we agree that giving the key aspects in this paper, and all specific initial and boundary conditions provides clarity. Therefore, we have added more detail on how the velocity and temperature perturbations were added in the forced case. After looking at the experimental particle bombardment events (Supplementary Movie 2 or 3), surface standing waves are typically generated for some time periods after the initial bombardment effect settles down. Therefore, we added standing wave perturbations on the surface in this forced case. Since this calculation case is to rationalise the experimentally observed oscillatory bubble motion, the standing sinusoidal wave characteristics are taken from the experimental observation. The wavelength λ is one-fifth of the longitudinal melt pool length (5 standing waves in the melt pool), the period T is 0.6 ms, the displacement amplitude A is 30 μm , and the velocity amplitude is $A\omega$, where $\omega = 2\pi/T$. In the assumed region of particle bombardment, the surface temperature is set at 1800 K, which may vary the Marangoni effect, but this temperature effect may be relatively minor compared to the velocity perturbations given as above. The original Marangoni flow patterns are changed to smaller cells. The results of these simulations demonstrate that the bubble oscillation can be a result of the surface standing wave generation due to particle bombardment.

This standing wave generation can also give rationalisation to the fact that the bubble motion is near-regular although particle bombardment is random (as commented by Reviewer #3). It should be noted that in the actual experiment, the unsteady generation of initial short ripple formation just after bombardment will make the bubble oscillation not as simple as the above assumption (λ is varying with time) but the surface wavy motion and the subsequent flow cell formation is the likely cause.

Please see the revision on pages 38-39 and below.

[Lines 718 - 725]: “In the forced case in Fig. 8, velocity and temperature perturbations were directly applied to the melt pool surface. From the experimental observation, standing waves are seen after particle bombardment. For simplicity, the perturbations on the surface are set as follows (based on the experimental observations): the wavelength λ is one-fifth of the longitudinal melt pool length (5 standing waves in the melt pool), the period T is 0.6 ms, the displacement amplitude A is 30 μm , and the velocity amplitude is $A\omega$, where $\omega = 2\pi/T$. In the assumed region of particle bombardment, the surface temperature is set at 1800 K, but this temperature effect is minor.”

7. In Fig.8 b and e, the bubbles cannot be seen in the simulation, while the bubble in fig. 8d is not as big as that observed in Fig. 8a.

Many thanks for pointing out what was a rendering error in making the figure. Specifically, in Fig. 8 b, d & e, there were no resolved bubbles added and only the point-particle bubbles in the fluid flow field were simulated (resolving bubble volume significantly slows the simulation), as our purpose was to demonstrate the change in flow on the impact of a particle, disrupting the Marangoni flow by local quenching and reducing the temperature gradient. This approach can be justified for small bubbles considered here. In Fig. 8d, a white blank dot seen in the figure is actually not a bubble but is caused by the threshold error in the gas-phase blanking operation in the post-processing software. We have modified the figure correcting our error. We thank the Reviewer for this comment. Please see the revised Fig. 8 below or on page 33.

[Lines 600 - 619]: “Fig. 8 Comparison between experimental data and modelling results of bubble migration in the melt pool.”

8. How well (in terms of the bubble sizes) was the solid/liquid interface trapping/pushing of bubbles reproduced in the simulation?

The bubble pushing / entrapment at the liquid/solid interface was not directly studied in the simulations used in this study, although the viscosity increase (and permeability reduction) in the mushy zone is considered and should be capable of simulating aspects of entrapment. This would be interesting future work, as this size of the bubble is affected by the locally different flow states around the bubble (for example, the upper-bubble flow and lower-bubble flow are not the same), and the flow difference is captured. At the final stage of the “bubble pushed by the flow” case (simulation result at $t = 4.5$ ms in Fig. 7b), the solid/liquid front has progressed toward the bubble, pushing the bubble slightly. This is indicated by the reduction in the $-z$ pressure force shown in Supplementary Table 3 (-7.21×10^{-7} N to -5.93×10^{-7} N), meaning that upward force is added by the solid/liquid front. This effect has been studied in a number of prior publications (e.g., Lee & Hunt, 2001¹⁴; Han & Hunt, 1994, 1995^{15,16}), instead our simulations focussed more on explaining the mechanisms of coalescence and why the large pores did not rise to the surface under the strong buoyancy forces, and the variation of Marangoni flow when particles impact the surface. Bubble capture is more evidently reproduced as the high viscosity and less permeability at the periphery around the solid/liquid front are included in the modelling, which causes the bubble's lower part to slow down and finally stick to the solid/liquid front (simulation result at $t = 4.5$ ms in Fig. 7b). The bubble shape is temporarily deformed due to the sheared upper part and the almost stopped lower part.

Reviewer #2 (Remarks to the Author):

In this paper, high-speed synchrotron x-ray imaging and multi-physics modeling were used to characterize the pore evolution mechanisms during the directed energy deposition process. The results are interesting and shed light on the pore minimization strategy development. Here are some comments for the authors to consider.

The authors appreciate positive comments and constructive suggestions for improving the manuscript. All comments have been addressed.

1. The travel speed of the substrate in Fig. 1 is 2 mm/s. I did a quick calculation and the laser-matter interaction time is 0.2 second. It may indicate a nearly symmetrical melt pool geometry with respect to the laser beam. It is suggested to provide the laser beam spot size definition, e.g., $1/e$, $1/e^2$, and confirm the laser beam locations (red color) in related X-ray images and schematics.

Thanks for your suggestions. The laser beam definition and laser beam locations are revised and clarified in the manuscript. Please see the revisions below or on pages 36, 12 and 11 in the revised manuscript and page 18 in Supplementary information. The laser locations in the corresponding figures, such as the Fig. 1 below, have been revised and confirmed.

[Lines 675- 677]: “The laser beam spot size is defined with $1/e^2$, and the profiled laser beam is plotted in Supplementary Fig. 14. The measured beam spot size is about 390 μm near the focal point.”

[Lines 171 - 173] in Supplementary information: “Supplementary Fig. 14. **a** The intensity colourmap of laser beam profiling. **b** the laser beam profile fitted by a Gaussian fitting. The white scale bar corresponds to 200 μm .”

[Lines 223 - 225]: “The laser beam in the X-ray radiographs and corresponding schematics are shown in red colour, and the laser beam location is nearly symmetrical to the melt pool geometry, while it is slightly in the forward of the centre due to the advection of heat.”

[Lines 210 - 226]: "Fig. 1 Dynamic bubble behaviour and mechanisms during DED."

2. In Fig. 2e, the lower travel speed of the substrate (1 mm/s) shows a much higher bubble formation rate from previous layers. It is largely attributed by the authors to the larger melt pool size. Related data like melt pool sizes could be provided.

We appreciate the suggestions. The melt pool sizes have been calculated and provided in the manuscript. Please see the revisions below or on page 17 in the revised manuscript and on page 10 in Supplementary information.

[Lines 330 - 334]: "The melt pool size at different traverse speeds, laser powers and layers are plotted in Supplementary Fig. 6. The melt pool length and depth are both larger at a lower speed, higher laser power and greater powder feed rate, while the layer effect is insignificant. This is related to the bubble growth behaviour as shown in Fig. 3 and Supplementary Fig. 5, *i.e.*, the larger melt pool allows the larger maximum bubble size reached."

[Lines 113 - 116] in Supplementary information: “Supplementary Fig. 6. Melt pool length and depth as a function of **a** traverse speed, **b** layer, **c** laser power and **d** powder feed rate. The error bars represent standard deviation. The inset in **a** shows the radiograph indicating the melt pool length and depth.”

3. Figs. 2e and 5c show an interesting trend in bubble number reduction. What would happen if the laser beam scans the same track without powder feeding for a second time or for many times? Will this help finally eliminate the bubbles?

Many thanks for suggesting this. In a recent study, we have already investigated what the reviewer suggests, and the reviewer is correct, most of the pores are removed, as shown in the image below. No changes to the text have been made as these results are part of a follow-on study not yet published.

As shown in the following Fig. R1, when the laser remelts the pre-build track, the pore marked with a green dashed circle is released from the solid into the melt pool (Fig. R1a-c) and then this bubble escapes (Fig. R1c-d). After remelting, the pores are mostly removed (Fig. R1d).

Fig. R1. The bubble removal with the laser remelting. **a**) the pore marked with the green dashed circle is entrapped in the solid at $t = 3000$ ms, **b**) the pore is released into the melt pool at $t = 3200$ ms, and **c**) the bubble marked with the yellow dashed circle moves at $t = 3304$ ms, and then **d**) the bubble escapes at $t = 3308$ ms. The laser power is 150 W, and the traverse speed is 1 mm/s. Scale bars are 300 μm .

Reviewer #3 (Remarks to the Author):

1. This work studies the pore evolution behavior during the directed energy deposition (DED) additive manufacturing process through a combination of in-situ X-ray imaging and multi-physics simulation. The pore formation and evolution during DED have been studied by in-situ X-ray imaging before. A representative work by S. Wolff et al. (International Journal of Machine Tools and Manufacture, volume 166, 2021, 103743, <https://doi.org/10.1016/j.ijmachtools.2021.103743>) using in-situ X-ray imaging with enhanced spatial and temporal resolution at the Argonne National Lab provided a detailed analysis of pore formation and evolution mechanisms during the DED process. Although Wolff's work used a small laser beam (which is more suitable for LPBF process instead of DED) to produce a keyhole mode melt pool, I don't see why some of their findings could not be applied to a conduction mode melt pool. For example, regardless of the melting mode the melt pool is experiencing, the pore formation from the "porosity inside the feedstock powder" should all be valid and expected.

The authors agree with Reviewer #3, many of the mechanisms documented by Wolff et al. ¹ might be applicable in a larger pool. However, our work significantly expands upon the work of Wolff et al. ¹, showing that:

1. Pores from the powder feedstock are the major source of porosity (Wolff et al. ¹ found <1% of the pores came from this source, termed "internal porosity" in their Fig 12);
2. Wolff et al.'s major source, gas entrained in front or behind the powder (termed 'entrapped pores'), was not observed at all with gas atomised powder and more industrial conditions;
3. Similarly, their other major sources, keyhole and interaction pores, were not observed in this study. Further, our work has many novel observations, and we use *in situ* observation and computational modelling as evidence for our hypotheses (as listed by Reviewer 1), namely that the coalescence of the small pores causes the formation of large, detrimental pores, a mechanism not previously reported.

We have now added further discussion to clarify where we observe similar mechanisms to those in Wolff et al. ¹, a comparison to our results, and where our results are different and novel. Please see the further discussion below or on pages 12-13 in the revised manuscript including:

[Lines 238 - 251]: "For the conditions used in this study, namely a gas atomised powder and conduction mode laser power, feedstock porosity is the dominant source of porosity. This was quantified by counting the newly formed pores over 100 ms of the build for each source (Fig. 2d), with the argon pores in the feedstock powder introducing 2 to 4 times as many as pores enter from all other sources. The only other source of bubbles we observed was those entering from the prior tracks (Fig. 2d). However, reference ⁵ suggested that during DED-AM of Ti-6Al-4V porosity can be generated from the feedstock, keyhole collapse, and by entraining gas when the powder particles enter the pool. For their conditions, using plasma atomised powders and laser conditions creating a keyhole, they concluded that feedstock porosity is a relatively insignificant contribution to the process with a contribution ratio of 0.22% ⁵. Our results show that for the more industrial conditions used here, feedstock porosity becomes the major source of pores rather than a negligible one. This would be the one of major differences in pore formation between this work and the prior study ⁵."

2. The pore evolution behaviors, as frequently stated in this paper, are strongly related to the Marangoni flow within the melt pool. However, the flow pattern in the conduction mode melt pool has been well studied before. For instance, the front and back outward recirculating flow cells were reported by L. Aucott et al. (Nature Communications, volume 9, 2018, 5414, <https://doi.org/10.1038/s41467-018-07900-9>), also using in-situ X-ray imaging. Similarly, Q. Guo et al. (Additive Manufacturing, volume 31, 2020, 100939, <https://doi.org/10.1016/j.addma.2019.100939>) revealed the side flows in conduction mode melt pool via in-situ X-ray imaging. With these groundworks, it is unsurprising that the majority of bubbles observed in this work should move in accordance with the flow patterns in the melt pool.

We agree with the Reviewer that the Marangoni flow is well known, observed by Mills et al.¹⁷ experimentally, and modelled by Paul & Debroy¹⁸ and Lee et al.¹⁹ over 3 decades ago, including using *ex situ* observation, and more recently by others including Aucott et al.²⁰ and Guo et al.²¹ who quantified it *in situ*. Our observations in DED are novel in that we show that some small bubbles follow the flow, some float out, some are entrapped, and some coalesce; whilst the large bubbles stay at the back of the melt pool – this has not been reported before. These novelties have been further highlighted as described in responses above, and the discussion revised as below or on page 35:

[Lines 649 - 654]: “The bubble behaviour should be related to the Marangoni flow in the melt pool. The Marangoni flow was observed by Mills et al.⁵⁸ and Lee et al.⁶¹ using *ex situ* observations, and modelled by Paul & Debroy⁶², and more recently *in situ* observations by Aucott et al.⁶³ for welding and Guo et al.⁶⁴ in LPBF. However, our observations in DED also elucidate that some small bubbles follow the flow, some float out, some are entrapped, and some coalesce; whilst the large bubbles stay in the melt pool.”

3. The paper also lacks in-depth analysis. For instance, in the section “Bubble migration mechanisms,” the content is merely a description of experimental observations and measurements, lacking analytical insight. If there is no analysis, then what is described here should not be referred to as “mechanisms”, but merely “behaviors”. The same concern applies to the section on “Bubble coalescence.”

Many thanks for your suggestion. The analysis has been enhanced both quantitatively and via increased discourse and comparison to the small amount of prior work in DED.

For example, please see the enhanced analysis for bubble migration below or on pages 19-20, and in additional discussion we’ve added to the text the responses to reviewers above on several other topics, and significant additions to the supplementary for analysis supported via modelling.

[Lines 364 - 367]: “In region A (Fig. 4b), which is the front of the melt pool, the bubble is observed to circulate counter clockwise, and the maximum velocity is measured to be ~88 mm/s, driven by Marangoni flow in the front of the melt pool.”

[Lines 377 - 379]: “When the bubble finally moves into region C, which is the back of the melt pool, its circular motion is observed to be clockwise, and its maximum velocity is ~196 mm/s, driven by the Marangoni flow in the rear of the melt pool, as shown in Fig. 4d.”

Please see the enhanced analysis for bubble coalescence below or on page 27.

[Lines 496 - 502]: “Bubble coalescence is much more likely to occur in the larger melt pool of DED than in LPBF, as the residence time of bubbles is much greater, enabling them to coalesce to form large bubbles. The strong recirculating flow in a large pool constrains both the small and large bubbles’ flow, creating conditions appropriate for bubble collision, with coalescence occurring when the film of liquid between colliding bubbles ruptures⁶⁰. Coalescence reduces the overall free energy as it minimises the total bubble surface area⁶⁰.”

4. In the section “Bubble pushing at the surface”, why the quantified numbers of the shear force and buoyancy force were not provided? Why was the hydraulic pressure not considered in the calculation? How does the shear force offset the buoyancy force, given that the shear force acts in the tangential direction of the bubble surface?

Many thanks for your suggestions for improvement. We have now included the calculation of the ratio of pressure force, shear force and buoyancy force. The bubble trajectory is a result of the balance of these forces including their acting directions. We have added descriptions in the main manuscript and supplementary information.

The discussion about the forces for bubble pushing was added below or on page 28 in the main manuscript. The full details can be found in Supplementary Discussion 2.

[Lines 515 - 522]: “In Supplementary Discussion 2, the force balance onto the large bubble is calculated by comparing static buoyancy, shear and pressure forces induced by the molten metal flow. According to the corresponding simulation results of these forces in Supplementary Table 3, the large horizontal shear force can push the large bubble in the horizontally backward direction. The strong transverse Marangoni flow above the bubble pushes the bubble downward, balancing the buoyancy and positive shear force in the vertical direction. Therefore, the bubble can be pushed downward and backward when this flow structure is formed.”

Supplementary Discussion 2 was added to enhance the discussion about the force calculation for the bubble. Please see Supplementary Discussion 2 on pages 20 - 21, and corresponding Supplementary Fig. 12 on page 16 and Supplementary Table 3 on page 23 in Supplementary information, respectively.

[Lines 192 - 224] in Supplementary information:

“Supplementary Discussion 2

The force balance onto the bubble is measured by calculating the static buoyancy, the shear force by the molten metal flow and the pressure force by the molten metal flow. They are given as

$$\text{Static buoyancy:} \quad (12)$$

$$\text{Shear force on the surface:} \quad (13)$$

$$\text{Pressure force on the surface (toward the bubble centre):} \quad (14)$$

where ρ_m is the molten metal density, ρ_g is the gas density, g is the gravitational acceleration ($= 9.8 \text{ m/s}^2$), V is the bubble volume, A is the bubble surface area, μ is the molten metal viscosity, \mathbf{v}_t is the tangential flow velocity vector just above the bubble surface, \mathbf{v} is the flow velocity and n represents the normal direction outward of the bubble.

\mathbf{e}_z is the unit vector pointing vertically upward, \mathbf{e}_n is the unit surface-normal vector pointing outward of the bubble and \mathbf{e}_t is the unit surface-tangential vector (projected along with the flow velocity direction).

From the simulation result, the forces are calculated as shown in Supplementary Table 3. The force direction and magnitude are illustrated in Supplementary Fig. 12c, d, g and h and Supplementary Fig. 13c and d. The main flow effects are shown as the blue arrows.

In the case of “Large bubble pushed” (Fig. 7b), the large horizontal shear force above the bubble is mainly pushing the bubble in the horizontally backward ($-x$) direction (the blue arrow in Supplementary Fig. 12c). At the same time, the transverse Marangoni flow (the blue arrow in Supplementary Fig. 12d) pushes the bubble vertically downward ($-z$). The positive shear force in the $+z$ direction is mainly due to the rising flow in the $+x$ side of the bubble (while the local shear force is toward the $-z$ direction in the $-x$ side of the bubble), but this is minor. In total, the obliquely descending flow above the bubble surpasses the bubble rising effect due to buoyancy. This strong flow strip above the bubble is due to the Marangoni effect on the melt pool surface, which supplies energy (work) to the flow. Therefore, the bubble is pushed backward and downward when this flow structure is formed. It should be also noted that the shear force causes the bubble to rotate in the clockwise direction as schematically shown in Supplementary Fig. 12. This bubble rotation can also be confirmed in Supplementary Movie 2. Note that the flow is unsteady and the quantitative magnitude of the forces may vary temporally, but the basic mechanism of pushing the bubble remains for some time. At 4.5 ms later (Supplementary Fig. 12g and h), the force analysis indicates that the bubble is still pushed by

the flow in the same direction. At this time, the bottom of the bubble is in the solidifying front region and the bubble is captured and will remain inside the melt pool.”

[Lines 159 - 164] in Supplementary information: “Supplementary Fig. 12. Modelling results showing bubble pushing. **a** 3D view and **b** top view of a large bubble in a rear/back location. **c** side view and **d** rear view of a large bubble pushed in melt pool at $t = 0.7$ ms from bubble insertion $t = 0$ ms. **e** side view and **f** rear view of bubble pushed in melt pool at $t = 1.9$ ms. **g** side view and **h** rear view of bubble trapped in solidification front at $t = 4.5$ ms. (Scale bars are $300 \mu\text{m}$). Force direction and magnitude are indicated with red arrows in **c**, **d**, **g** and **h**.”

[Lines 243 - 244] in Supplementary information: “Supplementary Table 3. Force magnitude on the large bubble”

Case		+x direction (horizontally forward)	+y direction (toward the centre of the melt pool)	+z direction (vertically upward)
Large bubble pushed	Buoyancy	-	-	1.50×10^{-7} N
	Shear force	-5.48×10^{-7} N	0.29×10^{-7} N	1.15×10^{-7} N

	Pressure force	-0.16×10^{-7} N	-2.41×10^{-7} N	-7.21×10^{-7} N
	Total force	-5.64×10^{-7} N	-2.12×10^{-7} N	-4.56×10^{-7} N
Large bubble pushed (4.5 ms later)	Buoyancy	-	-	1.40×10^{-7} N
	Shear force	-4.72×10^{-7} N	1.47×10^{-7} N	3.14×10^{-7} N
	Pressure force	-2.78×10^{-7} N	-6.81×10^{-7} N	-5.93×10^{-7} N
	Total force	-7.49×10^{-7} N	-5.33×10^{-7} N	-2.79×10^{-7} N
Large bubble pop up	Buoyancy	-	-	1.48×10^{-7} N
	Shear force	-0.49×10^{-7} N	-1.75×10^{-7} N	6.37×10^{-7} N
	Pressure force	-8.35×10^{-7} N	-2.58×10^{-7} N	0.69×10^{-7} N
	Total force	-8.84×10^{-7} N	-4.33×10^{-7} N	7.06×10^{-7} N

5. In the section “Large bubble escape”, further force analysis is required to support the explanation provided in Line 490: “breaking the balance of forces.” Moreover, large bubble escape is not limited to this one circumstance. For example, in Supplementary Video 1, some large bubbles disappeared suddenly inside the melt pool, without touching the top liquid surface; in Supplementary Video 2, some escaping bubbles were not in the middle of the melt pool, but in the rear circulation. None of them are explained in the paper.

We thank the Reviewer for the suggestions. Further force analysis was conducted to support the explanation regarding “breaking the balance of forces”. This is related to the previous comment, and we have modified it to include the calculation of the ratio of pressure force, shear force and buoyancy force. Similarly as in the previous case of the bubble pushed, the bubble escape case can also be understood by the (breaking of) force balance where the upward force is large now.

And with regards to the bubble appearing to pop with reaching the top in Supplementary Movie 1, this was in the 1 kHz imaging, which was not fast enough to capture motion to the top, as discussed below (see updated [Lines 534 - 542] below).

For clarity, we have added the following discussion on page 29 in the main manuscript or below. The full details can be found in Supplementary Discussion 2.

[Lines 530 - 533]: “Computational fluid dynamics simulation in the Supplementary Information (e.g., see Supplementary Fig. 13), show how changes in the Marangoni driven flow cells can create conditions entrapping bubbles within the flow cell, or pushing them to the melt pool surface, rupturing.”

Please see the corresponding Supplementary Discussion 2 on pages 20 - 21 and Supplementary Fig. 13 on page 17 in Supplementary information.

[Lines 225 - 234] in Supplementary information: “In the case of “Large bubble pop up” (Fig. 7c), due to the different locations where the bubble exists, the flow above the bubble is thinner and the shear force in the horizontally backward ($-x$) direction is weaker (the blue arrow in Supplementary Fig. 13c) and rather the front side flow pushes the bubble backward. More evidently, the transverse Marangoni cell flow (the blue arrow in Supplementary Fig. 13d) is much stronger at this x position (close to the melt pool centre in the x direction) and pushes the bubble vertically upward ($+z$) mostly by the shear force. Therefore, combined with the buoyancy, and with the fact that the distance between the bubble and the surface is closer, the bubble soon pops up to the melt pool surface and ruptures.

From the above results, it can be said that the structure of the Marangoni flow cells and the relative location of the bubble are the important factors to determine the bubble dynamics.”

[Lines 165 - 170] in Supplementary information: “Supplementary Fig. 13. Modelling results showing bubble pop up. **a** 3D view and **b** top view of a large bubble in a side location. **c** side view and **d** rear view of a large bubble pushed in melt pool at $t = 0.8$ ms from bubble insertion at $t = 0$ ms. The corresponding schematic of force directions in the bubble pop up case. Force direction and magnitude are indicated by the red arrows in **c** and **d**. **e** side view and **f** rear view of bubble pop up at $t = 0.96$ ms. (Scale bars are $300 \mu\text{m}$).”

The authors appreciate the reviewer for the suggestion. The bubble escape section has been enhanced. Please see the revision on page 29 in the main manuscript or below.

[Lines 534 - 542]: “Most bubbles escape through the top liquid surface of the melt pool, it requires the high-speed X-ray imaging with a frame rate of 20 kHz to capture these phenomena (see videos in Supplementary Movies 2, 3, 5 and 6), as the X-ray imaging at a low frame rate of 1 kHz may miss a short escaping period due to the fast bubble escaping speed of 247 mm/s in Fig. 5. A large bubble also escapes in the rear of the melt pool (see Supplementary Movie 2). The large bubble in the rear of the melt pool grows close to the top liquid surface of the melt pool, and the powder particle hits the melt pool and disrupts the Marangoni flow near the large bubble to break the force balance, so the large bubble can escape.”

6. In the section “Influence of powder particles hitting the melt pool surface”, what is the relative strength of impact wave versus temperature perturbations on the influence of bubble migrations? Which one is dominating, and under which circumstances? In addition, particle-hitting events are supposed to occur randomly in terms of impact location and timing. Why could such a random phenomenon possibly

cause a near-regular bubble movement in Figure 4c with a nearly fixed frequency and wavelength of the motion trajectory?

Thanks for pointing out this. For clarity, the section “influence of powder particles hitting the melt pool surface” has been enhanced accordingly. Please see pages 32 - 33 or below. (Also please see the response to Reviewer #1’s 6th comment.)

[Lines 582 - 599]: “When the powder particle hits the melt pool, it can mainly generate two effects, namely, (i) the impact ripple waves of the particle when the powder particle just touches the melt pool surface and subsequent standing waves, which can affect the flow and bubble migration near the particle; and (ii) after that, the powder particle gradually melts and quenches the melt pool, which can change the local temperature and flow pattern and bubble migrations near the particle. As shown in Fig. 8a-c and Fig. 4, the modelling results considering velocity and temperature perturbations for the powder effects are consistent with experimental results, in which the impact wave of powder particle causes the initial flow disruption and small flow cells (in accordance with the standing wave generation) are formed (Fig. 8d-f).

The motion trajectory in Fig. 4c and Fig. 8c is supposed to be mainly related to the simultaneous effects of Marangoni flow cells and powder impact effects. Although the powder particles can hit different locations of the melt pool at different times, the powder flow rate is controlled to be constant and high, which can produce a relatively consistent powder hitting, thus, to change the flow pattern in the melt pool. It is also speculated from the experimental results that the later standing wave formation is nearly similar although the initial ripple formation and the temperature effect occur in random places. Therefore, the bubble motion trajectory exhibits an organised pattern.”

7. With the above questions arising, I am not convinced that the work reported here has been properly interpreted, taking all details and scenarios into consideration.

We thank the Reviewer for their critical analysis and thought-provoking comments, but we feel that we have made novel observations, developed hypotheses of the underlying mechanisms that can explain these phenomena, and then used experimental and computational evidence to support them. The corresponding results and discussions have been enhanced and further interpreted according to the Reviewer’s suggestions. We are confident that our work will open new avenues for the study of pore evolution in DED. We appreciate that there is always more that could be done, but further work is outside the scope of this study.

8. The writing in the paper lacks organization and can be challenging to follow. For example, lines 145-153 are not relevant to pore formation and should be relocated. Descriptions in lines 382-385, 487-491 are overly vague and do not convey valuable information. Lines 525-528 are difficult to comprehend and could benefit from clarification or rephrasing.

Thanks for pointing this out. The corresponding revisions have been made to enhance the organisation of the full paper. For original lines 145 - 153, the descriptions have been relocated to section bubble coalescence and growth which are related to the flow. Please see the revision on pages 7-8 or as follows.

[Lines 157 - 165]: “In Fig. 1a-d, the outward Marangoni flow is expected to occur in the melt pool, as the surface temperature is the highest under the laser than near the edge of the melt pool, so the liquid flows from this low surface energy area out to the colder edges (higher surface energy) to minimise the free energy. This creates a very fast strong surface flow outward from the laser, creating recirculation flow cells at the front and back⁵⁸. Bubbles are observed to follow the outward Marangoni flow in the melt pool. Based on the 2D projections of the events (Fig. 1c), in the front and back regions of the melt pool ($t = t_0 + 146$ ms and $t_0 + 150$ ms), bubbles are observed to recirculate, driven by the Marangoni flow.”

For original lines 382 - 385, the description has been enhanced to clarify the valuable information. Please see the revision on pages 21-22 or below.

[Lines 401 - 407]: “Bubbles will escape if the buoyancy force is greater than the downward component of the recirculating cell. Another important factor, we hypothesise, is the location and velocity of the bubble inside the recirculation cell, as this also affects the upward component of the bubble, which ranges from 88 mm/s (Fig. 4b) to 247 mm/s (Fig. 5a) when the bubble changes from recirculation mode to escape. This indicates that the maximum bubble velocity in the vertical direction will affect bubble motion and hence escape.”

For original lines 487 - 491, the statement has been enhanced to clarify the valuable information about large bubble escape. Please see the revision on pages 28 - 29 or below.

[Lines 523 - 542]: “**Large bubble escape.** Fig. 7c (and Supplementary Fig. 13) shows an example where a very large bubble can escape from the top of the melt pool. The large bubble touches the top liquid surface when the bubble grows into a critical size and moves by the flow disruption, and then the top liquid surface ruptures to release the large bubble. Here the bubble is both very large (and hence large buoyancy force) and is located in the middle of the melt pool, between the flow recirculation cells, breaking the balance of forces, so the bubble pops up, explaining the experimentally observed behaviour. Computational fluid dynamics simulation in the Supplementary Information (e.g., see Supplementary Fig. 13), show how changes in the Marangoni driven flow cells can create conditions entrapping bubbles within the flow cell, or pushing them to the melt pool surface, rupturing.

Most bubbles escape through the top liquid surface of the melt pool, it requires the high-speed X-ray imaging with a frame rate of 20 kHz to capture these phenomena (see videos in Supplementary Movies 2, 3, 5 and 6), as the X-ray imaging at a low frame rate of 1 kHz may miss a short escaping period due to the fast bubble escaping speed of 247 mm/s in Fig. 5. A large bubble also escapes in the rear of melt pool (see Supplementary Movie 2). The large bubble in the rear of the melt pool grows close to the top liquid surface of the melt pool, and the powder particle hits the melt pool and disrupts the Marangoni flow near the large bubble to break the force balance, so the large bubble can escape.”

For original lines 525 - 528, the statement about bubble migration has been clarified and enhanced, please see the revision on page 31 or below.

[Lines 575 - 581]: “In addition, in the modelling results shown in Fig. 8e, an outward flow cell forms near hitting particles. In Fig. 8e and f, these flow cells can drive bubbles to migrate from the front to the rear of that melt pool (in the region indicated with a red dashed box) and then circulate outward (in the region indicated with a black dashed box). These phenomena are consistent with the experimental results of bubble migration in Fig. 8f and Fig. 4d. These results indicate that the flow cells generated by the particle impact can promote the bubble migration.”

References

1. Wolff, S. J. *et al.* In situ X-ray imaging of pore formation mechanisms and dynamics in laser powder-blown directed energy deposition additive manufacturing. *Int. J. Mach. Tools Manuf.* **166**, (2021).
2. Zhu, X. *et al.* Prediction of melt pool shape in additive manufacturing based on machine learning methods. *Opt. Laser Technol.* **159**, 108964 (2023).
3. Zhang, Y. M., Lim, C. W. J., Tang, C. & Li, B. Numerical investigation on heat transfer of melt pool and clad generation in directed energy deposition of stainless steel. *Int. J. Therm. Sci.* **165**, (2021).
4. Zhang, P. *et al.* Effects of melt-pool geometry on the oriented to misoriented transition in directed energy deposition of a single-crystal superalloy. *Addit. Manuf.* **60**, 103253 (2022).
5. Amine, T., Newkirk, J. W. & Liou, F. Investigation of effect of process parameters on multilayer builds by direct metal deposition. *Appl. Therm. Eng.* **73**, 500–511 (2014).
6. Ma, M., Wang, Z. & Zeng, X. A comparison on metallurgical behaviors of 316L stainless steel by selective laser melting and laser cladding deposition. *Mater. Sci. Eng. A* **685**, 265–273 (2017).
7. Song, J. *et al.* Numerical and experimental study of laser aided additive manufacturing for melt-pool profile and grain orientation analysis. *Mater. Des.* **137**, 286–297 (2018).
8. Zhai, Y., Galarraga, H. & Lados, D. A. Microstructure, static properties, and fatigue crack growth mechanisms in Ti-6Al-4V fabricated by additive manufacturing: LENS and EBM. *Eng. Fail. Anal.* **69**, 3–14 (2016).
9. Fang, J. X. *et al.* The effects of solid-state phase transformation upon stress evolution in laser metal powder deposition. *Mater. Des.* **87**, 807–814 (2015).
10. Kong, Y., Zhao, L., Zhu, L. & Huang, H. The selection of laser beam diameter in directed energy deposition of austenitic stainless steel: A comprehensive assessment. *Addit. Manuf.* **52**, 102646 (2022).
11. Kies, F. *et al.* Defect formation and prevention in directed energy deposition of high-manganese steels and the effect on mechanical properties. *Mater. Sci. Eng. A* **772**, 138688 (2020).
12. Anderson, I. E., White, E. M. H. & Dehoff, R. Feedstock powder processing research needs for additive manufacturing development. *Curr. Opin. Solid State Mater. Sci.* **22**, 8–15 (2018).
13. Ahn, D. G. *Directed Energy Deposition (DED) Process: State of the Art. International Journal of Precision Engineering and Manufacturing - Green Technology* vol. 8 (Korean Society for Precision Engineering, 2021).
14. Lee, P. D. & Hunt, J. D. Hydrogen porosity in directionally solidified aluminium–copper alloys: a mathematical model. *Acta Mater.* **49**, 1383–1398 (2001).
15. Han, Q. & Hunt, J. D. Particle pushing: critical flow rate required to put particles into motion. *J. Cryst. Growth* **152**, 221–227 (1995).
16. Han, Q. & Hunt, J. D. Particle pushing: the attachment of particles on the solid-liquid interface during fluid flow. *J. Cryst. Growth* **140**, 406–413 (1994).
17. Mills, K. C., Keene, B. J., Brooks, R. F. & Shirali, A. Marangoni effects in welding. *Philos. Trans. R. Soc. A Math. Phys. Eng. Sci.* **356**, 911–925 (1998).
18. Paul, A. & Debroy, T. Free surface flow and heat transfer in conduction mode laser welding. *Metall. Trans. B* **19**, 851–858 (1988).
19. Lee, P. D., North, T. & Perrin, A. R. Methods of experimental confirmation of a computational model of the fluid flow in gas tungsten arc welding. *Modeling and Control of Casting and Welding Processes. IV* 131–140 at (1988).
20. Aucott, L. *et al.* Revealing internal flow behaviour in arc welding and additive manufacturing of metals. *Nat. Commun.* **9**, 1–7 (2018).
21. Guo, Q. *et al.* In-situ full-field mapping of melt flow dynamics in laser metal additive manufacturing. *Addit. Manuf.* **31**, 100939 (2020).

REVIEWER COMMENTS

Reviewer #1 (Remarks to the Author):

The authors have addressed most of my previous comments. A last piece of suggestions before acceptance is to explicitly specify the limitation of the model. For example, the authors should explicitly specify that this model does not really resolve bubbles and its impact on molten pool flow to address the previous comment #7.

Reviewer #2 (Remarks to the Author):

The authors have reasonably addressed the points I provided. I would like to suggest an acceptance in its current form.

Reviewer #3 (Remarks to the Author):

I appreciate the authors' detailed response to my previous comments. However, my primary concern about the manuscript's limited novelty remains unchanged. I acknowledge that the authors have observed new phenomena, particularly the formation of large pores through the coalescence of smaller ones. Other findings, such as identifying powder-borne pores as a significant source of porosity, seem to be merely varying outcomes stemming from different processing conditions between the two studies, while the underlying pore formation mechanism is consistent.

Regarding bubble migration, it is indeed that not all bubbles follow Marangoni flow. The authors have listed four scenarios that deviate from this pattern: (1) bubbles floating out; (2) bubbles becoming entrapped; (3) bubbles coalescing; and (4) large bubbles remaining within the melt pool. For (1) and (2), the fate of the bubbles—whether they float out or become entrapped—is primarily determined by the interplay of thermocapillary, buoyancy, and drag forces, as detailed in S. Hojjatzadeh's work [Nat Commun 10, 3088 (2019). <https://doi.org/10.1038/s41467-019-10973-9>]. Regarding the large bubbles staying in the melt pool, I am not at all surprised. When I see the large vortex at the rear end of the conduction mode melt pool in the works of Aucott (<https://doi.org/10.1038/s41467-018-07900-9>) and especially Guo's (<https://doi.org/10.1016/j.addma.2019.100939>), it would be strange if it could not trap a pore.

Overall, I consider this to be a thorough and compressive study, but regrettably, I do not find the level of novelty and scientific advancement presented to meet the threshold required by Nature Communications. This assessment holds particularly when the work is compared to the representative articles published in related fields in Nature Communications in the last two years, such as <https://doi.org/10.1038/s41467-022-30667-z>, <https://doi.org/10.1038/s41467-022-28694-x>, and <https://doi.org/10.1038/s41467-022-28649-2>.

Detailed Response to Reviewers' Comments

Ref. No.: NCOMMS-23-16900A

Title: Pore evolution mechanisms during directed energy deposition additive manufacturing

Dear Reviewers:

The authors appreciate the time and effort that you dedicated to review this work. Many thanks for providing your valuable comments, which helped us to improve this manuscript significantly. The point-by-point response to these comments along with the substantial revisions made are listed below. Our responses are noted in blue text. The revised parts in this manuscript are shown below the responses (and as they are small are not highlighted in the main text).

Reviewer #1 (Remarks to the Author):

The authors have addressed most of my previous comments. A last piece of suggestions before acceptance is to explicitly specify the limitation of the model. For example, the authors should explicitly specify that this model does not really resolve bubbles and its impact on molten pool flow to address the previous comment #7.

Response: The authors appreciate the reviewer's positive and constructive comments to improve our manuscript. We agree with the reviewer that we had not sufficiently explained the two methods used for tracking bubbles (Lagrangian particle tracking for small bubbles, and level set VOF explicit simulation for the large bubbles). I.e., the small bubbles are not explicitly resolved but the large ones are. The main text has been revised, and a clear explanation and references have been added to Supplementary Method 1. The following paragraph has been added to the main text:

[Lines 715 - 720]: "For the small bubble tracking cases in Fig. 8b and 8d-e, these bubbles are assumed to be sufficiently small that they can be treated as Lagrangian point particles (see Supplementary Method 1 and Supplementary Fig. 15 for justification). For the large bubbles (e.g., those in Fig. 7), the bubbles are explicitly modelled using the level-set method to capture the liquid gas interface, simulating the surface shape and bubble coalescence (see Supplementary Method 1)."

And the following details have been added to Supplementary Methods:

[Lines 64 - 89] in Supplementary information: "For bubbles, two tracking methods are used, Eulerian interface tracking and Lagrangian point particle tracking. The appropriate regime of bubble modelling can be clarified by the particle Reynolds number, the particle Weber number and the Stokes time scale. The Reynolds number is given by

(8)

where ρ is the liquid density, D is the bubble diameter and μ is the liquid viscosity. U is the slip velocity magnitude between the bubble and the surrounding liquid flow. The Weber number is given by

(9)

where σ is the surface tension coefficient. The Stokes time scale is given by

(10)

which gives the relaxation time scale for the bubble to follow the outer liquid flow (namely, the slip velocity becomes negligibly small). The point particle assumption is justified when the

flow around the bubble is very slow and symmetric ($Re < 1$, for example) and the Stokes drag law can be used (τ is smaller than the flow time scale) and the bubble shape is spherical due to sufficiently stronger surface tension than the inertial force ($We < 0.1$, for example). Taking the lower limits of Reynolds number of 1.0, the Weber number of 0.1 and the Stokes time scale of 0.1 ms (from the experimental observations), the regime map becomes as shown in Supplementary Fig. 15. The region below each line represents the point particle justification for each parameter and the bottom-left region (green region) satisfies all the conditions. The small bubbles tracked in the experiment, which are mostly pre-existing bubbles from the feedstock, lie in this regime of the map and the point particle assumption is justified for these bubbles in the simulation (the method is described in the next paragraph). However, for the coalesced large bubbles, the slip flow velocity is ~ 0.3 - 0.4 m/s (max) and they are out of the point particle regime. In this case, the numerical modelling needs to resolve the shape of a finite-volume bubble, and Eulerian interface tracking by Eq. (7) is used.”

[Lines 201 - 204] in Supplementary information: “Supplementary Fig. 15. Bubble (pore) modelling regime. The green region in the bottom-left is the Lagrangian point particle regime. The experimentally observed large (coalesced) bubbles are in the yellow region where Eulerian interface tracking is needed.”

Reviewer #2 (Remarks to the Author):

The authors have reasonably addressed the points I provided. I would like to suggest an acceptance in its current form.

Thank you for your support.

Reviewer #3 (Remarks to the Author):

I appreciate the authors' detailed response to my previous comments. However, my primary concern about the manuscript's limited novelty remains unchanged. I acknowledge that the authors have observed new phenomena, particularly the formation of large pores through the coalescence of smaller ones. Other findings, such as identifying powder-borne pores as a significant source of porosity, seem to be merely varying outcomes stemming from different processing conditions between the two studies, while the underlying pore formation mechanism is consistent.

Response: The authors appreciate the reviewer's comments. We agree that our work reports new phenomena and findings including pore coalescence and formation mechanisms in additive manufacturing. In addition, our work also significantly expands the new pore evolution mechanisms in directed energy deposition (DED) including pore motion, pushing, entrainment and entrapment with both *in situ* high-speed X-ray imaging and multiphysics modelling.

Regarding bubble migration, it is indeed that not all bubbles follow Marangoni flow. The authors have listed four scenarios that deviate from this pattern: (1) bubbles floating out; (2) bubbles becoming entrapped; (3) bubbles coalescing; and (4) large bubbles remaining within the melt pool. For (1) and (2), the fate of the bubbles—whether they float out or become entrapped—is primarily determined by the interplay of thermocapillary, buoyancy, and drag forces, as detailed in S. Hojjatzadeh's work [Nat Commun 10, 3088 (2019). <https://doi.org/10.1038/s41467-019-10973-9>]. Regarding the large bubbles staying in the melt pool, I am not at all surprised. When I see the large vortex at the rear end of the conduction mode melt pool in the works of Aucott (<https://doi.org/10.1038/s41467-018-07900-9>) and especially Guo's (<https://doi.org/10.1016/j.addma.2019.100939>), it would be strange if it could not trap a pore.

Response: The authors appreciate the reviewer's comments, and we are aware of the prior work they highlight. The first paper (Hojjatzadeh et al., Nat. Commun. ¹) studies the migration of small bubbles during laser powder bed fusion (LPBF) using X-ray radiography, supported by a heat and mass transfer model that does not include any bubble simulation. Our work is the first work to report the bubble migration and evolution in directed energy deposition (DED), which has significantly different length and driving force scales, and hence mechanisms compared to the previously published LPBF studies. For example, the melt flow and pore velocities in LPBF ¹ are reported as 600-2000 mm/s, which is 10 times of the pore velocity in our DED work of 20-250 mm/s. Hojjatzadeh et al. ¹ provide an approximate analytic estimate of the relative thermocapillary, buoyancy and drag forces for these quite different LPBF conditions, which are several orders of magnitude different from DED pool velocities/pore sizes. Our multiphysics model explicitly calculates these relative forces (for large explicitly modelled bubbles) showing that these larger bubbles will remain in the pool until nearly reaching the size of the recirculating flow, when the melt flow forces at the bottom of the pore start acting opposite to those on the top.

In terms of Aucott et al. and Guo et al.'s studies, we agree that they also show there is a strong flow vortex in arc welding ² and LPBF ³ respectively; however, neither studied pore motion and coalescence, although as the reviewer suggested, their experiments both show recirculating flows but do not report or discuss the impact of these flows on the large coalescing pores we observe in DED. To the best of our knowledge, our work is the first study in additive manufacturing to report the large bubbles formed by dozens of coalescing small bubbles (from the feedstock) that stay in the melt pool; further we hypothesise the underlying mechanisms and validate these with *in situ* X-ray imaging and multiphysics modelling.

Overall, I consider this to be a thorough and compressive study, but regrettably, I do not find the level of novelty and scientific advancement presented to meet the threshold required by Nature Communications. This assessment holds particularly when the work is compared to the representative articles published in related fields in Nature Communications in the last two years, such as

<https://doi.org/10.1038/s41467-022-30667-z>, <https://doi.org/10.1038/s41467-022-28694-x>, and <https://doi.org/10.1038/s41467-022-28649-2>.

Response: We thank the reviewer for their thought-provoking comments. However, we feel that our work has made significant advancements and novelty in the additive manufacturing field, particularly in DED. The previous work cited by the reviewer focuses on the phenomena in LPBF including the vapour jet ⁴, keyhole fluctuation ⁵ and spatter ⁶. Our work is the first comprehensive research to study bubble evolution in DED at the time scale of seconds, compared to milliseconds in LPBF. Therefore, we feel that our work is a significant contribution to the field of AM, particularly DED.

References

1. Hojjatzadeh, S. M. H. *et al.* Pore elimination mechanisms during 3D printing of metals. *Nat. Commun.* **10**, 1–8 (2019).
2. Aucott, L. *et al.* Revealing internal flow behaviour in arc welding and additive manufacturing of metals. *Nat. Commun.* **9**, 1–7 (2018).
3. Guo, Q. *et al.* In-situ full-field mapping of melt flow dynamics in laser metal additive manufacturing. *Addit. Manuf.* **31**, 100939 (2020).
4. Bitharas, I. *et al.* The interplay between vapour, liquid, and solid phases in laser powder bed fusion. *Nat. Commun.* **13**, 2959 (2022).
5. Huang, Y. *et al.* Keyhole fluctuation and pore formation mechanisms during laser powder bed fusion additive manufacturing. *Nat. Commun.* **13**, 1170 (2022).
6. Qu, M. *et al.* Controlling process instability for defect lean metal additive manufacturing. *Nat. Commun.* **13**, 1–8 (2022).

REVIEWERS' COMMENTS

Reviewer #1 (Remarks to the Author):

The authors have addressed my previous comment